# Creating a psychosocial intervention combining growth mindset and implementation intentions (GMII) to reduce alcohol consumption: A mixed method approach

**Sacha Parada**[1]⊛*, **Bérengère Rubio**[2]⊛, **Elsa Taschini**[3‡], **Xavier Laqueille**[3‡], **Malika El Youbi**[4‡], **Pierre Paris**[4‡], **Bernard Angerville**[5‡], **Alain Dervaux**[5‡], **Jean-François Verlhiac**[1]⊛, **Eve Legrand**[1]⊛

**1** Parisian laboratory of social psychology (*LAPPS*), University Paris Nanterre, Nanterre, France, **2** Clinical, psychanalitical and developmental laboratory (*CLIPSYD*), University Paris Nanterre, Nanterre, France, **3** Addictology department, Sainte-Anne Hospital, Paris, France, **4** Addictology department, Dreux Hospital, Dreux, France, **5** University Paris-Saclay/EPS Barthélémy Durand, Etampes, France

⊛ These authors contributed equally to this work.
‡ MEY, PP, BA and AD also contributed equally to this work.
\* sacha.parada@gmail.com

**Data Availability Statement:** Data regarding this study is available here: https://osf.io/z6yhv/.

## Abstract

This work aimed at creating a psychosocial intervention based on growth mindset theory and implementation intention strategies, in order to reduce alcohol consumption among users in the general population, and the clinical population of individuals with alcohol use disorder. A mixed method approach was used, combining qualitative and quantitative research methods among both populations. Four focus groups were first conducted to extract arguments in favor of a malleable view of alcohol consumption (study 1A), situations that trigger the desire to drink alcohol, as well as strategies used by people to counteract this need (study 1B). Data were analyzed using reflective thematic analysis in line with the scientific literature on alcohol consumption. The results were used to create a questionnaire scoring the relevance of each argument, situation and strategy (study 2). The 20 best scored arguments, situations and strategies were selected to create the intervention. The created intervention consisted in a popularized scientific article describing alcohol consumption as malleable, including the selected arguments and followed by two internalization exercises. Then, a volitional help sheet included the selected situations and solutions was presented, allowing forming up to three plans. The discussion focused on the added value of the created material compared to pre-existing tools in the literature, and presents plans to test the intervention in a future study.

**Funding:** SP, EL, and JFV were funded by the National Cancer Institute for this research (Institut National du Cancer—INCa_16214. https://www.e-cancer.fr/). The funders had no role in study design, data collection and analysis, decision to publish, or preparation of the manuscript.

**Competing interests:** The authors have declared that no competing interests exist.

## 1. Introduction

Theory-based psychosocial interventions [1] have been applied to a variety of health topics such as physical activity, weight management, mental health (anxiety, depression, stress, etc.) or substance use disorders. The purpose of these interventions was to foster benefic behavioral changes in people. The present work aimed to create a new psychosocial intervention in order to reduce alcohol consumption for the clinical population of patients with Alcohol Use Disorder (AUD), and alcohol users among the general population. The intervention combined existing behavior change theories and strategies, namely growth mindset theory (GM—[2]) and implementation intentions (II—[3]). This type of intervention, labeled GMII (i.e., Growth Mindset and Implementation Intentions), was original and thought to be able to foster motivation and changes among individuals [4]. The present GMII intervention was created using both empirical qualitative and quantitative data and input from the scientific literature.

### 1.1. Alcohol, risks and prevention in the international and French populations

According to the World Health Organization [5], 3 million deaths every year were attributable to harmful use of alcohol worldwide, representing 5.3% of all deaths, and 13.5% of all deaths among people aged 20 to 39 years. Harmful use of alcohol was a causal factor in more than 200 diseases and injury conditions. Around 1.3% of the world population (more than 100 million people) were diagnosed with an AUD in 2016 [6]. In Europe, alcohol was the second cause of preventable mortality, and was prevalent in France with approximately 49000 deaths imputable to alcohol yearly. It represented a social cost estimated at 118 billion euros in 2010 in France, for a total of 42,8 million drinkers with varied consumer profiles [7]. It was proven to increase the risk of physical and psychological illnesses, such as cancer [8] and depression [9]. Around 86% of the French population had been drinking alcohol in 2017 [10], 40.1% on a frequent and regular basis (one to three times a week), and at least 10% of the population was drinking every day. Twenty-three percent of the French population would have a hazardous consumption, and 7% an AUD [11].

In France, the prevention of alcohol consumption aimed to avoid and/or delay the consumption of the first drink and avoid and/or reduce problematic or intense consumption. Prevalently, mass prevention was marked by the targeting of specific, high risk populations (e.g., young people, vulnerable older people [12]). Designing and conducting interventions remained costly (e.g., 10,5 billion had been devoted to prevention in 2002 in France–including all prevention domains–[13]), and multiplying interventions in order to target specific populations increased this cost even more. Despite this focus on high-risk users, recent data show that alcohol consumption in France has a quasi-normal or normal distribution [7], meaning that most drinkers have an average level of consumption, rather than an extreme one. Consequently, in order to reduce excessive consumption, it would be possible to target a reduction in the average consumption of the population rather than targeting specific groups, thereby achieving an overall reduction in consumption. The first aim of the present project was then to construct a single intervention able to foster changes in drinking behavior among different types of consumers (from occasional drinkers to individuals with AUD), by targeting two processes thought to be common among them: implicit theories of alcohol consumption, and consumption automatisms.

The second aim of the project was to implement empirically tested and complementary behaviour change strategies. Indeed, prevention measures were deemed more likely to be effective when based on theories and models of behaviour change that have been documented, correctly implemented and evaluated [7]. In this idea, Michie and al. [1] developed a taxonomy

indexing 93 behavior change techniques, grouped under 16 clusters (e.g., goals and planning; reward and threat; self-belief; etc.). Furthermore, they introduced the behavior change wheel [14] to help identify and apply these techniques when targeting a specific change in individuals. At the center of the wheel is the COM-B model (Capability, Opportunity, Motivation– leading to Behavior). It posits that it is possible to encourage change by acting on the different sources of behavior, namely individuals' abilities (e.g., increasing knowledge about alcohol and its risks); opportunities (e.g., creating opportunities to carry out another activity instead of drinking alcohol); and motivation (e.g., encouraging the desire to change one's consumption). Another model, the transtheoretical model of change [15] complements the COM-B by proposing stages of change and strategies for moving through these stages. In the precontemplation stage, the individual does not think they have a problem with the product, and therefore does not want to change their behaviour. In the contemplation stage, they become aware of the problem but are still reluctant to change their consumption. In the preparation stage, they are motivated and ready to change their behaviour. In the action stage, the change in behaviour is effective. This is followed by the maintenance stage, in which the new behaviour is consolidated over time. Finally, there is the relapse stage, which is an integral part of the behavioural change process. The present project aimed to act on most sources of behavior and phases through the mindset theory and implementation intention strategies.

## 1.2. Growth mindset and alcohol outcomes

Mindset theory posits that individuals can endorse two distinct perceptions of their internal characteristics (e.g., intelligence, personality, tendency to drink alcohol). They can hold either a fixed mindset, believing the targeted trait is fixed and cannot be changed or improved upon; or a growth mindset, believing they can always change and grow [2]. Endorsing a growth mindset would be beneficial for individuals in health contexts (e.g., protection against setback-related weight gain [16]; protection against psychological distress following stressful life events [17]) compared to a fixed mindset. Consequently, interventions have been used to improve mental health outcomes. Schleider et al. [18] used a self-administered, single session, computerized GM intervention to successfully reduce depressive symptoms in rural adolescents. It included different modules related to different types of mindsets (personality, intelligence, self-regulation), containing scientific information and peer testimonies on why ability in a given domain has the potential to change, as is the norm in the mindset literature [16, 19, 20]. Participants then completed a "saying is believing" exercise aimed at better internalizing the message, by allowing deeper cognitive processing [21].

From the behavior change taxonomy of Michie and collaborators [1], GM interventions would be categorized in the "shaping knowledge" cluster, and the technique "4.3 –reattribution" (i.e., elicit perceived causes of behavior and suggest alternative explanations, such as external or internal and stable or unstable). In the COM–B model of the behavior change wheel [14], GM would act on psychological capability and reflective motivation (i.e., understanding and believing one can change). When considering the transtheoretical model [15], GM would help individuals navigate through the first three stages of change, namely precontemplation (i.e., not wanting to change), contemplation (i.e., being aware that a problem exists and thinking about overcoming it), and preparation (i.e., wanting to change and preparing to act). Indeed, it could reinforce the belief that they can change themselves and their problematic behavior (i.e., drinking) or make them realize that they might have a problematic consumption.

Few studies linked growth mindset to alcohol outcomes (for a review see [22]). Most studies were correlational and showed that the growth mindset generally appeared to be negatively

linked to alcohol use [23–27]. Studies which linked mindset and addiction outcomes (including alcohol abuse) focused on a wide range of mindsets (e.g., addiction mindset, intelligence mindset, morality mindset, substance abuse mindset, drinking tendencies mindset, alcoholism mindset, etc.). The link between mindset measures and substance use outcome appeared stronger when they matched (i.e., smoking mindset and smoking [28]; drinking tendencies mindset and alcohol abuse [26]), compared to unrelated mindsets such as intelligence or morality [29] as well as a more holistic mindset such as general mental-health mindset [26]. As a result, the present work focused on alcohol consumption mindset, rather than AUD [25] or drinking tendencies mindsets [26]. Thus, it matched as closely as possible the behavior targeted by the GMII intervention, namely reducing alcohol consumption.

To our knowledge, no interventional growth mindset study had been conducted on alcohol consumption [22]. In a related work on addiction, Sridharan et al. [30] created a web-based intervention in order to decrease tobacco addiction. They created six lessons to counteract 6 beliefs about the permanence of nicotine addiction, based on scientific evidence in favor of the malleability of the targeted concept (i.e., nicotine addiction). Participants in the intervention group had higher smoking cessation rate compared to the control group. Another experimental study [23] created a GM message aimed at improving addiction outcomes. They presented a "compensatory-growth" message (vs. a "disease-fixed" message) in the form of a *Psychology Today* type article describing the many potential causes of drug and alcohol addiction, highlighting the potential to change or offset the condition in the future. Following the text, participants had to summarize the main theme of the article in one sentence, rate its understandability for 9[th] graders, and offer suggestions to improve it. The authors observed positive effects on perceived self-efficacy, intention to pursue counseling and treatment related to alcohol and drug use. In sum, mindset theory and interventions have been used to improve health outcomes, and more recently substance use disorders, with encouraging yet scarce results [22]. To improve the potential to foster change in alcohol consumption, it appeared appropriate to mobilize another tested behavior change technique, namely implementation intentions [3, 31].

## 1.3. Implementation Intentions and Alcohol Outcomes

Implementation Intentions (II) are planning strategies that aim to increase goal achievement [3]. They consist in planning when, where and how to act in the future in order to achieve a desired goal. More precisely, individuals have to anticipate a good opportunity to act (e.g., "If I am thirsty") or an obstacle to goal achievement (e.g., "If I am tempted to drink alcohol"). Situations can advantageously be intern (e.g., feelings, thoughts, physical states) or extern to individuals (e.g., moments, locations). Then, they plan how to respond to the anticipated situation: how to take advantage of the opportunity or how to avoid or overcome the obstacle. Responses can be behavioral (e.g., "Then I will order water"), cognitive (e.g., "Then I will tell myself I can resist") or emotional (e.g., "Then I will stay calm and relax"). Finally, the chosen situation and response have to be linked in the specific format "If situation X, then response Y" (e.g., "If I am tempted to drink alcohol when I am at the bar, then I will order a lemonade"). The efficacy of II on goal achievement pertains to two processes. First, following the formation of this strategy, the "If" situation is highly accessible in memory [32–34]. Second, the "Then" behavior is endorsed automatically when the situation is encountered, meaning quickly, efficiently, with little awareness and controllability (for a review see [35]).

A recent meta-analysis [36] showed that 16 studies used implementation intentions in relation to alcohol outcomes. Implementation intentions have been used to successfully reduce binge drinking [37] and alcohol consumption [38–40], and so among students [41, 42], women with moderate consumption [43] and the general population [44]. Online II protocols

have been used effectively [45]. The effect size difference for forming implementation intentions on weekly alcohol consumption was $d_+ = -0.14$, $CI$ [−0.24; −0.03]. In the behavior change taxonomy [1], II would express the technique "1.4 –action planning" in the "goals and action planning" cluster. In the COM-B model [14], II would act on social and physical opportunity, reflective motivation (i.e., planning a behavior change in a concrete situation), and automatic motivation once the new healthy behavior is adopted (i.e., change in drinking habits). From the transtheoretical model perspective [15], II could allow individuals to plan the change and enact it when the situation arises and maintain it over time, effectively transitioning from preparation to action (i.e., modifying the behavior to overcome the problem) and maintenance (i.e., maintaining the new behavior over time) stages. Implementation intentions were shown to help people move through the stages of change [46].

Among the various interventions that applied II to alcohol outcomes, a specific tool called "volitional help sheet" (VHS) seemed particularly promising [39, 44, 47] and would be more effective in reducing weekly alcohol consumption compared to the classic implementation intention format [36]. The VHS listed a set of critical situations and goal-directed responses (e.g., reducing alcohol consumption) in two parallel columns. Critical situations reflected temptations participants might experience in their daily life (e.g., "If I am tempted to drink more than the government recommends when I am excited" [44]). Appropriate responses were originally "processes of change", or means by which behavior might be changed, drawn from the transtheoretical model [48] (e.g. "Then I will avoid situations that encourage me to drink" [44]). Participants were asked to identify situations where they might be tempted to drink, and to link them to appropriate responses (relevant to them personally) by drawing lines between critical situations and appropriate responses, then write the resulting "If. . . Then. . ." plans [39, 44, 47]. This design was thought to facilitate the formation of implementation intentions, by helping individuals to identify relevant situations and responses, and make plans of good quality by allowing to clearly link them in an "if-then" format.

## 1.4. Growth mindset and implementation intentions, a new type of psychosocial intervention

GM and II were deemed complementary when considered in a behavior change framework, such as the COM-B model [14] and the transtheoretical model of change [15]. In the COM-B model, GM and II could act on the main different sources of behavior to foster change (i.e., capability, opportunity, and motivation). From the transtheoretical model perspective, GM could help transition from precontemplation to contemplation, to preparation, whereas II would lead individuals from preparation to action and maintenance stages. In previous work, the authors [4] showed the effectiveness of an intervention combining growth mindset and implementation intention (GMII) to foster a growth mindset of intelligence among university students. This intervention was more effective than the GM only condition, and its effects lasted longer. By combining GM and II, the purpose of this work was to maximize the potential for change.

## 1.5. Purpose of the studies

To design a GMII intervention able to foster changes in drinking behavior, it appeared pertinent to target both the general population of drinkers, and the clinical population of individuals with AUD. This was done to reflect the diversity of individuals affected by alcohol consumption, with varying degree of severity. Furthermore, implicit beliefs about alcohol consumption, as well as situations and strategies pertinent to users, could greatly vary depending on their profile. By focusing on both the clinical and general population, the intervention

would account for this diversity. Considering the lack of literature for GM interventions on substance and alcohol use disorders [22], as well as the specificity of the targeted populations, using empirical data in addition to the scientific literature appeared relevant to design the intervention. To this end, a mixed approach was used. Firstly, in study 1, with a qualitative focus, focus groups were conducted [49] with reflexive thematic analysis [50] to extract arguments promoting a malleable view of alcohol consumption (study 1A); as well as critical situations that might arouse the desires or habits to drink alcohol, and appropriate responses that people might use in those situations in order not to drink or drink less (study 1B). Secondly, in study 2, adopting a quantitative method, each argument, situation and strategy collected during the focus groups, as well as inputs from the scientific literature, were scored (study 2). The best ranked items were finally used to create the GMII intervention. As the intervention's objective was to reduce alcohol consumption among the general and clinical populations of alcohol consumers, each study's sample was taken from both populations.

## 2. Study 1

Study 1 used a qualitative approach, and four focus groups were conducted in the general and clinical populations to collect arguments in favor of a malleable view of alcohol consumption (study 1A), situations that arouse the desire to drink, and responses that individual use not to drink or drink less in such situations (study 1B). Study 1 follow the SRQR (Standards for Reporting Qualitative Research) guidelines for reporting qualitative research [51]. To reach the study's objectives, a narrative research approach was used with reflexive thematic analysis, in an interpretivist paradigm.

### 2.1. Researcher characteristics and reflexivity

The researcher conducting the focus groups (SP) had experience with qualitative research method and with the population. He knew the objectives of the groups and expected to find arguments, situations and solutions related to the study's goal. He had no previous or future relationship with participants. Clinical participants shared their experiences in a context similar to that of a support group, which might have caused expectations. The researcher made sure to clarify that the groups did not have a therapeutic objective and were not substitute for medical care.

### 2.2. Data collection and analysis

Audio data was collected from the four focus groups. A professional audio recorder was used. Recordings were transcribed, coded and anonymized by a researcher (SP) and his student assistant. Themes, subthemes and components were first identified using reflexive thematic analysis [50] by SP, then discussed by SP, BR, EL, and JFV for several iterations until an agreement was met. This approach allowed for flexibly generating themes, subthemes and components around alcohol consumption, while keeping in mind the objectives of the focus groups to extract arguments related to GM, and situations and responses related to II.

### 2.3. Participants and recruitment

For study 1A, eight participants were recruited from the general population (six women and two men) to form the first group. Ages ranged from 35 to 67 years old ($M = 53.1$, $SD = 12.3$). Eight participants were recruited from the clinical population (five women and three men) to form the second group, ages ranged from 43 to 64 years old ($M = 53.6$, $SD = 7.0$). Mean AUDIT scores in the clinical group were $M = 25.6$, $SD = 6.5$ for women and $M = 18.0$, $SD = 1$

**Table 1. Participants characteristics.** M = man; W = woman.

| Study | Group | Participant | Sex | Group | Age | AUDIT Score | Socioprofessionnal category |
|---|---|---|---|---|---|---|---|
| 1A | Clinical | $P_1$ | W | C | 43 | 29 | Employee |
| | | $P_2$ | W | C | 45 | 19 | Executive |
| | | $P_3$ | W | C | 55 | 31 | Executive |
| | | $P_4$ | W | C | 64 | 31 | Retired |
| | | $P_5$ | W | C | 60 | 18 | Executive |
| | | $P_6$ | M | C | 55 | 17 | Executive |
| | | $P_7$ | M | C | 54 | 18 | Intermediate profession |
| | | $P_8$ | M | C | 53 | 19 | Executive |
| | General | $P_9$ | W | G | 67 | 5 | Retired |
| | | $P_{10}$ | W | G | 55 | 3 | Employee |
| | | $P_{11}$ | W | G | 45 | 4 | Intermediate profession |
| | | $P_{12}$ | W | G | 59 | 2 | Employee |
| | | $P_{13}$ | W | G | 61 | 3 | Employee |
| | | $P_{14}$ | W | G | 35 | 4 | Business manager |
| | | $P_{15}$ | M | G | 38 | 2 | Intermediate profession |
| | | $P_{16}$ | M | G | 65 | 5 | Intermediate profession |
| 1B | Clinical | $P_{17}$ | W | C | 43 | 33 | Executive |
| | | $P_{18}$ | W | C | 50 | 31 | Other (artist) |
| | | $P_{19}$ | W | C | 31 | 28 | Executive |
| | | $P_{20}$ | M | C | 57 | 26 | Employee |
| | | $P_{21}$ | M | C | 48 | 24 | Employee |
| | | $P_{22}$ | M | C | 45 | 34 | Executive |
| | | $P_{23}$ | M | C | 72 | 8 | Retired |
| | General | $P_{24}$ | W | G | 32 | 1 | Employee |
| | | $P_{25}$ | W | G | 36 | 5 | Unemployed |
| | | $P_{26}$ | M | G | 51 | 1 | Executive |
| | | $P_{27}$ | M | G | 71 | 6 | Executive |
| | | $P_{28}$ | M | G | 80 | 1 | Retired |

for men. For study 1B ($N$ = 12), five participants were recruited from the general population (three women and two men) to form the first group. Ages ranged from 32 to 80 years old ($M$ = 54, $SD$ = 21.1, one participant did not report their age). Seven participants were recruited from the clinical population (two women and four men) to form the second group, ages ranged from 31 to 72 years old ($M$ = 49.4, $SD$ = 12.7, see Table 1). Mean AUDIT scores in the clinical group were $M$ = 30.7, $SD$ = 2.52 for women and $M$ = 23, $SD$ = 10.9 for men. All individual scores were higher than the cutoff of 13 suggested for alcohol dependence in the French population [52]. Sampling stopped when an appropriate number of participants for each focus group was reached [49]. The number of groups was limited to four due to time, funding and recruitment constraints.

Participants from the general population were recruited using an online recruitment platform (i.e., the French Risc list). After having been informed of the study, they emailed the researcher, who sent them the eligibility questionnaire. If eligible, a date was set to participate in the study. Participants were welcomed into an experiment room at University Paris Nanterre set for group discussions and signed an informed consent. The discussions lasted about 1.5h. Participants were then debriefed, their questions answered, and they reported their demographic information. They were thanked and given a gift certificate worth 30€. Participants from the clinical population were recruited in the Sainte-Anne's Hospital *CSAPA* (center

for care, support and prevention of addictions) of Paris, France. After agreement with their referring physician, the researcher contacted the patient and proposed the study. The eligibility questionnaire was sent to volunteers, following which a meeting was set at the hospital. The focus groups' process was the same as for the general population, with the discussions lasting about 2h. The longer discussion time for the clinical population might be explained by the fact that clinical participants were more used to discuss in a group context. Indeed, some participants had already taken part in group therapy sessions at the time of recruitment. For the general population, participants might never have taken part in such a group discussion before, and were potentially less comfortable to share, or more reluctant to express personal ideas in public. This caused the exchanges to be shorter and resulted in shorter groups duration.

For eligibility, clinical participants must fulfill AUD CIM-10 criteria at the time of recruitment, current or in remission for less than a year. They must be considered able to undergo the study (assessed by their referring physician). Participants from the general population must not have been diagnosed with AUD and must not present hazardous drinking behavior (score < 7 for men and < 6 for women on the AUDIT questionnaire [52], see *Measures* section). All participants must consume alcohol (score > 0 on the AUDIT first item), must not present another substance use disorder (score < 5 on the DAST questionnaire) aside from smoking, must speak French fluently, be over eighteen and not be under guardianship. One participant scored 7 on the DAST but was still included in the clinical group of study 1A. After discussion, it appeared they did not correctly understand the measure and included their alcohol consumption in their response. All participants were recruited between February and April 2022. The principal investigator (SP) had access to information that could identify individual participants during and after data collection.

## 2.4. Materials and measures

**2.4.1. Discussion guides.** For study 1, two semi-structured discussion guides were used to conduct the focus groups. The discussion guides were pretested on a sample of university students (2 groups, $n_1$ = 9, $n_2$ = 8) to design questions and probes. The main questions were asked to students during the pretest, and the variety of their response helped to refine the probes to adequately cover the questions' subjects and elicit pertinent data, as well as improve their understandability. The discussion guides were the same for the general and clinical populations. Study 1A aimed to foster a discussion about the malleability of alcohol consumption, what one should think or do to change their consumption. Study 1B focused on collecting situations that might trigger the desire to drink alcohol, strategies that people might use to resist this desire, and potential obstacles they might encounter when implementing these strategies. The experimenter asked questions, and used prompts when needed to stimulate the conversations (see Table 2).

**2.4.2. Care for alcohol use disorder.** All participants reported whether they had received any sort of care (medical, psychological, others) for alcohol use disorder (AUD) according to the CIM-10 criteria. If positive, they indicated if they currently took medical treatment for AUD (if yes, for how long), another type of medical care for AUD (for how long), psychological care for AUD (for how long), if they have ever been hospitalized for AUD (how many times), if they were ever in rehab for AUD (how many times), if they ever tried to stop drinking alcohol (how many times), or have abstinence periods (how long it lasted). Participants from the general population pool recruited online (Risc list) that answered positively were excluded, as clinical participants were recruited exclusively from the hospital.

**2.4.3. Alcohol use and abuse.** The French version [52] of the Alcohol Use Disorders Identification Test (AUDIT—[53]) was used. It can be used to detect alcohol consumption, AUD

**Table 2. Focus groups discussion guides.**

| Study | Question | Probe |
|---|---|---|
| **Introduction** | *We want to know about people's perceptions of drinking alcohol, whether you think it's something that can be controlled, or not, and why* | |
| *1A* | **According to you, what is alcohol addiction?** | To what behaviors, attitudes, desires or emotions does it refer to? |
| | **Do you think one can control and positively change or evolve its alcohol consumption?** | What skills would one need to do so? |
| | | What techniques or exercises might exist to help change one's alcohol consumption? |
| | | What thoughts, reasoning or mindset one should have to positively change its alcohol consumption? |
| | | What should I tell someone to convince them they can change their alcohol consumption? What mindset should I instill in him, what way of thinking should he adopt? |
| | | On the other hand, what can lead the individual to think that they cannot change their alcohol consumption, that they are in some way doomed to remain an alcoholic? |
| | | What thoughts or situations might lead me to think that I can't change my drinking? |
| | **Can alcohol addiction be cured or healed? If so, how? What does one would need?** | Is alcohol addiction a disease? Is it more than that, or different? |
| | | What sets apart those who can get out of addiction and those who fail? Do you think some people are more predisposed than others to get out of addiction? In what way? |
| **Introduction** | *We want to identify situations that might arouse the need to drink alcohol, and strategies that one might use to not drink (or drink less) in such situations* | |
| *1B* | **According to you, what is alcohol addiction?** | To what behaviors, attitudes, desires or emotions does it refer to? |
| | **Are there any situations that triggers an urge to drink alcohol?** | Does being around certain people, being in certain places, experiencing particular emotions or events trigger a desire to drink alcohol? |
| | | Without mentioning a particular "craving," are there any situations where you have a habit of drinking alcohol? |
| | | Do you know of any other situations that might trigger people to drink? Imagine your friends and family: have you noticed situations (places, times of the day, emotions) that trigger a desire to drink? |
| | | What do you think makes it easier to access alcohol? |
| | **Have you ever implemented strategies or techniques to avoid drinking when you felt like it?** | Have you ever tried not to drink alcohol when you felt like it? And what did you do or tell yourself to do so? |
| | | Without necessarily having used them, have you ever heard of techniques that might help you not drink alcohol, or imagined things that you might not have done to avoid drinking alcohol or to drink less? |
| | | What would you tell a loved one who drinks alcohol to avoid or stop drinking, and what techniques would you suggest? |
| | | What other actions could be taken (in terms of prevention) to help a particular person (who is aware of the problem) to decrease or stop using alcohol? |
| | **What obstacles can prevent us from applying the strategies for not drinking alcohol when we want to?** | Have you ever tried not to drink alcohol when you wanted to? What prevented you from doing that?" |
| | | Without necessarily having experienced them, have you ever heard of barriers to not drinking alcohol, or imagined barriers that might arise when trying to avoid drinking alcohol or drink less? |
| | | What would you say to a loved one who drinks alcohol to help that person avoid drinking or stop drinking, what obstacles should they watch out for? |

and hazardous drinking. The cutoff scores set for hazardous drinking for the French version of the scale are $\geq 7$ for men and $\geq 6$ for women [52]. Cronbach's alphas for study 1A and study 1B were .92 and .95. respectively.

**2.4.4. Other drug abuse.** The 10-item version of the French-Canadian translation [54] of the Drug Abuse Screening Test (DAST-10 [55, 56]) was used to screen participants without a drug abuse or dependence problem. The French-Canadian was slightly adapted to improve its comprehensiveness. The terms designing "illegal drugs" and "prescribed medication" being distinct in the French language, "medication" was added to next to "drugs" in each item so as not to oversight medication abuse (e.g., "Have you used medication or drugs other than those

required for medical reasons?"; "Have you neglected your family because of your use of medication or drugs?"). Using a high cutoff score raises specificity and is more effective for screening of nondrug abusers [57]. Thus, while different cutoff scores are used across studies ($\geq 2$ [58], or $\geq 3$ [59]) the present studies used the cutoff score of $\geq 5$ (as it is possible to fine-tune it for an optimal selection in a particular context [60]).

**2.4.5. Sociodemographic questions.**   Participants were asked to indicate their age, sex, if French was their native language, guardianship status, socioprofessional category, highest degree, and average monthly household income.

## 2.5. Results

**2.5.1. Study 1A.**   Study 1A focused on belief participants held about the malleability of alcohol consumption. Two major themes emerged (see Table 3).

*2.5.1.1. Beliefs about alcohol consumption.* The first theme encompassed participants' beliefs about alcohol consumption. These beliefs first referred to a fixed mindset, such as thinking that changing is difficult at each stage: difficult to be aware that one's consumption is problematic (relates to the precontemplation stage), difficult to find motivation and to enact change (relates to the contemplation and action stages), because people have established habits of alcohol consumption (that are hard to break), to the point where drinking has become too difficult to control, to stop consumption, and has become an unstoppable need. Alcohol has been reported as enclosing individuals, removing their free will to choose not to consume. The social norms around alcohol, its acceptability and presence in society, maintain these beliefs engrained in individuals. The last component referred to description of alcohol as a disease. Individuals here reported that this disease acts at the neurological level (often perceived as impossible to change [23]) and that it may or may not be cured.

A second component was extracted, regrouping expressions of malleable beliefs about alcohol consumption. Several participants reported that it is possible to change mindset and adopt a positive way of thinking that favors change, and to change one's cognitions about alcohol. It is possible to believe in rebuilding oneself even after a period of problematic consumption, by having projects and improving one step at a time. Changes were seen favorably when feeling the benefits of stopping or reducing consumption, implying the idea that decreasing or stopping is possible. Participants also reported that it is possible to be abstinent from alcohol, partially or totally, even with a history of problematic use. Another clue of malleability pertains to the detailed strategies participants described in order to reduce their consumption (e.g., monitoring, replacing alcohol with something else, using social support).

*2.5.1.2. Determinants of alcohol beliefs.* The second theme regrouped the triggers of alcohol consumption and reduction and the consequences of drinking. Alcohol use can have different origins such as trauma or difficult life events. Everyday drinking can then be triggered by emotional states and in particular state-seeking behaviors such as disinhibition, anesthesia or pleasure; or by social factors such as pressure to drink or isolation. Finally, the participants developed different triggers for reducing their consumption, in particular medical (e.g., stroke) or social (e.g., request from the entourage). The second component refers to the consequences of consumption. They were differentiated in three areas: physiological (e.g., sleeping difficulties, skin and liver problems), psychological (e.g., causes anger, distress, shame) and social (e.g., desocialization, social categorization and reject). Finally, inter-individual differences were mentioned in terms of the risk of overconsumption and of developing harmful consequences. Two factors in particular were mentioned here, namely financial wealth and resistance to the effects of the product.

**Table 3. Emerging themes and components around the mindset of alcohol consumption according to FG (number of focus groups in which the component was mentioned) and *N* (number of participants having mentioned the component).**

| Themes | Subthemes | Components | Verbatim | FG | N |
|---|---|---|---|---|---|
| **Beliefs about alcohol consumption** | Fixed beliefs | *Habits of alcohol consumption* | "When I was teleworking, even, it was the period when from time to time actually I drank a little bit at lunch" | 2 | 12 |
| | | *Difficulties to find motivation and to enact change* | "I can't imagine eating oysters without wine or cheese." | 2 | 10 |
| | | *Difficulties to be aware of the problematic consumption* | "It's true that I was in denial for years, really, it was incredible" | 2 | 9 |
| | | *Difficulties to control consumption* | "I find it very difficult to stop at one drink" | 2 | 11 |
| | | *Alcohol encloses individuals* | "Free will does not work anymore. After a while, there is no more choice. After a certain number of drinks." | 1 | 4 |
| | | *Social norms of alcohol* | "At home, there was that in the culture. We were put in the head that alcohol was festive, that's it, but there was the culture, the culture of the vine." | 2 | 5 |
| | | *Alcohol as a disease* | "Unfortunately, there is always a rate of people who. . . [. . .] who, who are incurable, we will not be able to save it is sad to say. . ." | 2 | 8 |
| | Growth beliefs | *Changing mindset* | "The answer for me would be to tell myself [. . .] that I am not better off with alcohol." | 1 | 5 |
| | | *Believing in rebuilding oneself* | "Being able to make projects, for oneself with others, already I think it's putting oneself in a more positive state of mind" | 2 | 9 |
| | | *Feeling the benefits of stopping/reducing alcohol* | "It's difficult at the beginning [to stop drinking. . .] and then it's a liberation [. . .] psychologically and physiologically. it clears up a bit more in the brain" | 2 | 6 |
| | | *Beliefs in total or partial abstinence* | "I had a colleague who was a total boozer and smoked like crazy. [. . .] from one day to the next, I was stunned by the guy, not a drop of alcohol and not a cigarette." | 2 | 10 |
| | | *Strategies of change* | "If I set up at the time of the cravings, I put on my sneakers I go for a walk I also cook a lot" | 2 | 15 |
| **Determinants of alcohol beliefs** | Triggers of alcohol consumption and reduction | *Origins of alcohol consumption* | "It has links with things a little traumatic which took place" | 2 | 13 |
| | | *Emotional causes* | "[After drinking the first glass] Then I'm happy, everything's fine, there's the pleasure factor, I drink a lot of wine." | 2 | 10 |
| | | *Social causes* | "I think that people who don't have the problem [of alcohol], they don't realize at all, how complicated it can be for people who have it." | 2 | 9 |
| | | *Triggers for reducing consumption* | "At one point, I was drinking alcohol I said to myself: "attention, danger"." | 2 | 7 |
| | Consequences of alcohol consumption | *Physiological consequences* | "It's the physiological and physical consequences afterwards, of course, we feel them." | 2 | 11 |
| | | *Psychological consequences* | "But it's often related to depression, by the way [alcohol use]." | 2 | 12 |
| | | *Social consequences* | "It also de-socializes [alcohol] after socializing." | 2 | 7 |
| | | *Interindividual differences* | "People will act differently depending on whether they can handle it more or less. Some people can handle alcohol better than others." | 2 | 6 |

**2.5.2. Study 1B.** Study 1B focused on sampling a variety of critical situations in which people consume alcohol, and appropriate responses that they might use. Three major themes emerged (see Table 4).

*2.5.2.1. Triggers leading to alcohol consumption.* The first major theme focused on triggers and incentives that might lead people to consume alcohol. Participants from both groups gave social justification to their alcohol consumption. Triggers to alcohol consumption might be being with others (e.g., friends, family), wanting to integrate the group and do as others do (for the general population), avoiding marginalization (i.e., when others wonder why the participant does not drink and pressure them) or being alone (e.g., after work, when no one is

**Table 4. Emerging themes and components around critical situations and appropriate responses according to FG (number of focus groups in which the component was mentioned) and *N* (number of participants having mentioned the component).**

| Themes | Subthemes | Components | Verbatim | FG | N |
|---|---|---|---|---|---|
| **Triggers leading to alcohol consumption** | Social dimension | *Being with others* | "When I actually start drinking, it's excruciating it's. . . the cravings are triggered when I'm in society with others." | 2 | 6 |
| | | *Get into a group* | "We want to be a part of the group" | 1 | 5 |
| | | *Being marginalized* | "I think it can also have a bit of an offbeat look to it to say "well, but you refuse to drink how come?"" | 2 | 6 |
| | | *Being alone* | "Because if I stay alone too long, it's, it's part of the. . . the triggers." | 2 | 6 |
| | Life event | *Positive events* | "It was usually afterwards when we went to the restaurant, late at night [that we started drinking]" | 2 | 9 |
| | | *Negative life events* | "We knew very well that he was going to divorce, that he had lost everything a few, a few years ago, he had never recovered, on the contrary he had completely gone downhill. . ." | 2 | 7 |
| | Seeking. . . | *. . .A culinary pleasure* | "It also allows you to appreciate the dishes, it is not enough to drink without consuming at the same time [. . .]" | 2 | 6 |
| | | *. . .A new state* | "Afterwards, it's to be quiet, anaesthetized, cushy, all alone." | 2 | 7 |
| | | *. . . to diminish a negative state* | "And. . . well gradually [drinking], it was to alleviate situations of stress and solitude." | 2 | 10 |
| | Cognitions | *Lack of motivation* | "I think a lot, maybe sometimes too much, sometimes I'm also told that the will doesn't. . . it's not that it's enough" | 2 | 4 |
| | | *Permissive thinking* | "I am reasonable, why not have a little drink the next day. But always in moderation." | 2 | 4 |
| | | *Reactance* | "Alcohol must be a bit similar, the more people tell me to stop, the more I say "but no, it's my freedom, it's me, it's my choice" | 2 | 4 |
| | | *Unaware of one's consumption* | "Once you're drunk, you [. . .] don't count anymore! no, but that's not it, even I know I'm over the quota, but I go anyway so I. . ." | 2 | 4 |
| **Managing alcohol consumption** | Reasons to manage | *Acting on health consequences* | "And also, that's really on. . . in the short term, I tell myself no, I can't because, afterwards, I won't be able to sleep, because I won't be able to take the medication." | 2 | 8 |
| | | *Acting on daily life consequences* | [laughs] yeah or sometimes alcohol can loosen the tongue a bit and you don't pay too much attention to what you say, well I [. . .] yes but then sometimes with colleagues it's. . . that's it. . . I know that I can say things that I'll regret later. And the fact that I drank will make me. . . [. . .] that. . . | 2 | 8 |
| | | *Moment of realization* | "It is sometimes I thought "what am I doing, what am I saying?" The remarks of others, the family, etc." | 2 | 7 |
| | Strategies to manage | *Substitute* | "I always test all the soft drinks that exist." | 2 | 8 |
| | | *Avoid* | "I lock myself up at home [. . .] without anything [. . .] I know that there are no temptations." | 2 | 6 |
| | | *Monitor* | "It can be saying well no I don't drink today, I'll drink tomorrow" | 2 | 8 |
| | | *Social support* | "Never be alone for too long, always have a phone contact or see someone" | 2 | 9 |
| | | *Strategy sustainability* | "My personal situation has evolved there very recently, I have to find other strategies, the one that may have worked before, will not work either." | 1 | 2 |
| **Alcohol, individuals, and society** | Alcohol in society | *Accessibility of alcohol* | "I followed a professional training which led me to a job where people drank a lot too" | 2 | 10 |
| | | *Culture of alcohol* | "The day of the baptism we always put the finger, with a little rum, champagne, in the baby's mouth. When the baby is teething, the mother will rub the gums with rum" | 2 | 4 |
| | | *M/W differences* | "In some professional circles, a woman who drinks. . . it's less serious I think, these days." | 2 | 4 |
| | Causes and consequences of alcohol | *Origins and evolution of consumption* | "Sometimes you can also be addicted due to youthful consumption" | 2 | 8 |
| | | *Alcohol effects* | "Sometimes I managed to leave work early, because I couldn't control the tremors" | 2 | 10 |
| | | *Alcohol as a disease* | "It's recognized as a neurological disease, that's why we're here in fact." | 2 | 6 |

available). Positive daily events, such as partying or being on vacation, might also trigger alcohol consumption for both groups. Conversely, negative life events such as professional or personal problems (e.g., divorce) also influence consumption. Individuals might be in search of something specific that alcohol would bring, such as a culinary pleasure, though these reasons concern more the general population. Consumers might seek to attain a new state by drinking, such as being drunk or feeling relaxed. They might also want to diminish a negative state or feelings (e.g., stress, anxiety, sleeping problems). Finally, specific cognitions might trigger the need to consume alcohol, such as lacking motivation (e.g., having difficulties refusing a drink), having permissive thoughts (e.g., minimizing negative health consequences), being in reactance against not drinking alcohol when confronted with such discourse, or not being aware of one's consumption.

*2.5.2.2. Managing alcohol consumption.* The second theme focused on the reasons and the way participants manage their alcohol consumption. Participants invoked several reasons to manage their alcohol consumption. They fear the potential consequences of alcohol on their health (e.g., diseases, weight gain) and daily life (e.g., negative consequences with family and friends, being unable to function well the next day), and might have experienced a moment of realization driving them to change, be it by specific events (e.g., car crash) or relatives' intervention. They employ a variety of strategies in order not to drink or to drink less. These strategies can be of finding a substitute for alcohol (e.g., another drink, food, doing sports). Participants from both groups might also seek to avoid critical situations by not having alcohol at home, or going home to avoid temptations. They use strategies to monitor their consumption, like taking note of each drink consumed, or planning a set number of drinks as to avoid excess. Participants also report using social support, be it from peers, relatives, or medical professionals, to help them manage alcohol consumption. Two clinical participants also report having trouble finding sustainable strategies.

*2.5.2.3. Alcohol, Individuals, and society.* This last theme was more exploratory in nature, and investigated the different representations participants might have about alcohol, as well as its etiology. Alcohol is seen as easily accessible and present in society, in the daily personal and professional life of many. It is described as being engrained in the culture, starting with upbringing and education, and consumers being seen as socially desirable. Differences in norms and consumption between women and men are mentioned, with alcohol being more accepted for men than for women. Alcohol might have different origins (e.g., family, genetics), and alcohol consumption might evolve (i.e., increase) with time. Alcohol has numerous different effects on individuals (e.g., fosters negative emotions, adverse health outcomes, provides pleasure). Finally, it is seen as a disease mostly by the clinical participants. This disease is considered neurological, chronic, emotional and insidious.

## 2.6. Discussion

Focus groups analysis allowed to collect valuable qualitative data. Concerning beliefs about alcohol consumption, both fixed (e.g., difficulties to change lifestyles), and growth beliefs (e.g., changing mindset) emerged. Similarly, a wide range of situations that might trigger the desire to drink alcohol was discussed by participants. Situations can be cognitive (e.g., permissive thinking), contextual (e.g., being with others), or motivational (e.g., seeking to diminish a negative state). Several types of responses might be adopted by individuals to manage alcohol consumption, such as replacing, avoiding, monitoring, or using social support strategies. Themes discussed by participants echo what can be found in the alcohol literature. Concerning growth beliefs about alcohol consumption, we found beliefs that abstinence is necessary to manage alcohol abuse [61] implying a change in drinking behavior (related to "beliefs in total or partial

abstinence" component in the group 1A). People also believed it is possible to recover from alcohol and be perceived as functioning in society [62] (related to "believing in rebuilding one-self" component in the group 1A). It has been shown that reducing alcohol led to decreased physical and mental symptoms (for a review see [63], related to "feeling the benefits of stop-ping/reducing alcohol" component in the group 1A). Discussed fixed beliefs also relates to the literature, such as the role family consumption of alcohol plays in individuals' substance abuse [64] or that alcohol is an integrant part of certain life experiences such as college [65] (related to "social norms of alcohol" component in group 1A; and to "culture of alcohol" component in the group 1B). The brain disease model of addiction (BDMA) (related to "alcohol as a dis-ease" component in group 1A and 1B) is strongly supported by part of the scientific community [66, 67] and it was argued that a disease-centered view of addiction would necessarily lead to endorse fixed beliefs about addictions [22, 23]. Individuals also believed that alcohol can have adverse physiological consequences [68] (related "physiological consequences" component in the group 1A). Craving, or urges to drink alcohol is considered a potential predictor of relapse [69,70] (related to the component, "difficulties to control consumption" in the group 1A).

Concerning risks factors and what might lead individuals to alcohol consumption, negative life events [71] (related to "negative life events" component in the group 1B), permissive social norms (related to component "social norms of alcohol" in the group 1A and "accessibility of alcohol" in the group 1B) and wanting to unwind after work (related to "seeking a new state") were associated with risky alcohol use [72]. In agreement with participants' comments, alcohol supplied by the family has been shown to increase adverse drinking outcomes [73], similarly to peer influences [74] and social isolation [75] (related to "social dimension" theme in the group 1B and "social causes" component in groups 1A). Established contexts of alcohol consumption might also be a trigger to consume [76] (related to the "habits of alcohol consumption" compo-nent in the group 1A). Sensation seeking (related to « Seeking. . . » theme) [77], and negative psychological states (related to "emotional causes" in the group 1A and "seeking to diminish a negative state" component in the group 1B), such as anxiety disorders [78], negative affectivity [79, 80] or mental distress [71] are all linked to hazardous alcohol use. Concerning protection factors, social support (related to "strategies of change" component in group 1A and "social support" component in the group 1B) played an important part in preventing or reducing harmful alcohol use, as well as participating in social and leisure activities (related to "substi-tute" component in the group 1B) [81]. Behavior change techniques (BCT) seemed effective to improve alcohol consumption outcomes, such as self-monitoring [82] (related to "monitor" component in the group 1B). Other BCT appeared particularly promising to reduce alcohol consumption, such as "avoidance/reducing exposure to cues for behavior" (related to "avoid" component in the group 1B), and "pros and cons" (related to "determinants of alcohol beliefs" theme in the group 1A; and "reasons to manage" theme in the group 1B) [83].

Taken together, qualitative data collected in groups 1A and 1B was rich, and aligned with the literature on alcohol and addiction. Study 2 aimed to refine these findings with a quantita-tive approach to develop the GMII intervention.

## 3. Study 2

Study 2, with a quantitative method and inputs from the scientific literature, aimed to select the best arguments, situations and responses collected in study 1. To do so, an online study was conducted and participants from the general and clinical populations were asked to rank the rel-evance of each argument, situation and response on Likert-type scales. The goal was to select the 20 best ranked arguments, situations and responses (10 from the clinical population and 10 from the general population for each) and to use them to design the GMII intervention.

### 3.1. Participants and recruitment

For study 2 ($n$ = 97), 60 participants from the general population agreed to participate (46 women and 14 men). Ages ranged from 19 to 81 ($M$ = 40.7, $SD$ = 14.3). Thirty-seven participants from the clinical population were recruited (23 women and 14 men), ages ranged from 22 to 69 ($M$ = 44.9, $SD$ = 11). Participants who completed the entire survey dropped to 56 for the general population and 24 for the clinical population.

Participants from the clinical population were recruited via Facebook (Meta) groups dedicated to addiction and alcohol care, as well as the CSAPA, with the same recruitment method as study 1. Participants in the general population were recruited via the Risc list. The study consisted of a two-part online questionnaire (including the eligibility questionnaire) taking about an hour to complete in total. The first part contained the eligibility questionnaire (care for AUD, AUDIT, DAST) and the GM items to score. The second part of the questionnaire was sent the next day by email. It contained the II items to be scored. All participants were awarded a 20€ gift certificate once they fully completed both parts of the study.

The same eligibility criteria as in study 1 were applied for study 2, except for the physician's evaluation, as the study was done online. The same "care for alcohol use disorder" measure than in study 1 was used to discriminate between clinical and general participants. However, the hazardous drinking criteria for the general population was removed, meaning hazardous or dependent alcohol consumers (AUDIT score $\geq$ 7 for men and $\geq$ 6 for women) not diagnosed with AUD could participate in the study. Thus, the sample was more representative of the general population of drinkers, containing individuals with different degrees of severity in hazardous drinking. All participants were recruited between June and October 2022. The authors did not have access to information that could identify individual participants during or after data collection.

### 3.2. Measures

The measures of care for alcohol use disorder, alcohol and other drug use, and sociodemographic questions were the same as in study 1. Cronbach's alpha for the AUDIT scale was .92.

**3.2.1. Mindset of alcohol consumption.** Four items based on Schroder et al. [26] measured the mindset of alcohol consumption: "*Individuals have a certain degree of alcohol consumption, and they really cannot do much to change it*"; "*Alcohol consumption is something about oneself we cannot change very much*"; "*To be honest, we cannot really change our alcohol consumption*"; "*No matter how hard we try, we cannot really change our alcohol consumption*". Items were scored from 1 = *completely disagree* to 6 = *completely agree* on a Likert-type scale. Items were formulated so they expressed a fixed belief about alcohol consumption. This was done to avoid the potential social desirability bias of reporting malleable beliefs [25, 84], and is the standard in practice in mindset literature [26, 85]. The items were created using the "find-and-replace" method [26] consisting of using previously validated mindset measures, and replacing the original concept with the one of interest (e.g., switching "weight" with "alcohol addiction") [23, 25, 26]. Scores were reversed so that a high score indicates a rather growth mindset. Alpha for study 2 was .86 ($M$ = 4.23, $SD$ = 0.78).

**3.2.2. Agreement and convincingness of arguments for the malleability of alcohol consumption.** Study 1, as well as previous research on mindset and addiction [23] allowed for the collection of 27 distinct arguments in favor of a malleable view of alcohol consumption. These arguments were each transformed in one-sentence items and integrated into a Likert-type scale. For example, from the verbatim collected in group 1A (see Table 3): "*it's liberation [to stop drinking. . .] both psychologically and physiologically. It clears up a bit more in the brain [. . .] it's very beneficial*" two arguments were extracted: "*Successfully cutting down on alcohol*

 

*makes you feel better physically*"; and "*Successfully cutting down on alcohol makes you feel better mentally*". Participants first had to rate their agreement with each statement, from 1 = *completely disagree* to 6 = *completely agree*. They were then presented with the same items, and had to indicate how each proposal could help convince an individual to reduce their alcohol consumption, from 1 = *not convincing at all* to 6 = *very convincing*. Each item was then attributed a mean score for agreement and convincingness, as well as a third score averaging the two (for the 27 arguments: mean $r$ = .53, min $r$ = .41, max $r$ = .71, all $p$s < .001).

**3.2.3. Frequency of encounters and susceptibility to consume of critical situations.** Study 1 and literature search [44, 47] allowed for the collection of 62 distinct situations that might arouse a need to drink alcohol (employing the same method as described in the previous section) (see Table 5). For each item, participants rated the frequency to which they encounter them from 1 = *I never encounter this situation* to 6 = *I often encounter this situation*, as well as their own susceptibility to consume alcohol in such situations from 1 = *I never drink alcohol* to 6 = *I always drink alcohol*. Each situation was attributed mean frequency and susceptibility scores, as well as a third score averaging the two (for the 62 situations: mean $r$ = .52, min $r$ = .15, $p$ = .18; max $r$ = .88, $p$ < .001).

N.B. Arguments, situations and responses included in the final GMII intervention are in **bold**. Arguments marked * are based on [23]. Situations and responses marked * are based on [44, 47]. Situation 27, responses 23, 27, 14, and 22 have been modified in the final material to increase comprehension and specificity. Original items were "drinking out of habit"; "consult health professionals to talk about my alcohol consumption"; "surround myself with people who respect the right not to drink alcohol"; "don't have alcohol in the house to avoid being tempted to drink"; "avoiding situations that trigger the desire to drink alcohol".

**3.2.4. Effectiveness and self-efficacy of appropriate responses.** Study 1 and literature search [44, 47] resulted in establishing 47 distinct strategies that individuals might use to reduce (or stop) alcohol consumption (see Table 5). Participants were asked to rate the effectiveness of each item to reduce alcohol consumption from 1 "Would not allow me to reduce my alcohol consumption at all", to 6 "Would definitely allow me to reduce my alcohol consumption" and the perceived self-efficacy to apply the strategies from 1 "I do not feel capable of doing it at all" to 6 "I feel fully capable of doing it". A mean effectiveness and self-efficacy score were computed for each strategy, as well as a third score averaging the two (for the 47 responses: mean $r$ = .48, min $r$ = .27, max $r$ = .72, all $p$s < .015).

## 3.3. Statistical analysis

The Jamovi Software (v 2.3.18) was used to conduct the analysis. The between-subject variables were the population (clinical vs. general) and the gender (men vs. women). ANOVA, T-test and correlations were performed on AUDIT and mindset scores. Alpha level defining statistical significance was set at .05. Data regarding this study is available here: https://osf.io/z6yhv/?view_only=40ed1f2fb327462f842f83c5de102c19

## 3.4. Results

**3.4.1. Group differences and mindset of alcohol consumption.** A general linear model analysis was used to assess potential differences in AUDIT scores between men and women for the general and clinical populations $F(1,93)$ = 1.04, $p$ = .31. The interaction was not significant. However, AUDIT scores significantly differed between groups, $F(1,93)$ = 98.94, $p$ < .001, $\eta^2_p$ = .51, the clinical population scoring higher ($M$ = 25.0, $SD$ = 6.5) than the general population ($M$ = 9.0, $SD$ = 7.6), as might have been expected. To test the ability of the mindset scale to discriminate between the general and clinical populations' beliefs about the malleability of

 

**Table 5. Arguments for a malleable view of alcohol consumption, critical situations and appropriate responses extracted from the focus groups.**

| N° | Malleable arguments | Critical situations | Appropriate responses |
|---|---|---|---|
| 1 | **It is possible to cut down on your alcohol consumption** | **Being with family** | Plan my consumption in advance |
| 2 | **Cutting down on alcohol is a source of pride and should be congratulated** | **Being in a place where alcohol is consumed (bar, restaurant,...)** | Set a maximum number of drinks per occasion |
| 3 | **Cutting down on alcohol is a way to change yourself, to evolve, to become a different person** | **Being at a party** | Drink slowly |
| 4 | **To reduce your alcohol consumption, you have to do things step by step** | **Being on vacation** | **Enjoy quality rather than quantity** |
| 5 | To reduce your alcohol consumption, you must not take everything head on | **Being on a weekend** | **Refuse to drink alcohol** |
| 6 | **It is perfectly possible to do without alcohol, because alcohol does not make you more interesting** | **Being alone at home** | Keep a glass in my hand without drinking it |
| 7 | Stopping drinking does not lead to social rejection | Wanting to relax after work | **Order a non-alcoholic drink** |
| 8 | **It is not necessary to drink to be included in a group** | **Feeling like treating myself** | Eating instead of drinking alcohol |
| 9 | **It is possible to have a good time when you party without alcohol** | **Feeling like having a good time** | Playing sports instead of drinking alcohol |
| 10 | **Even a heavy drinker can successfully cut down on their drinking** | Feeling like being drunk | Meditate or do yoga instead of drinking alcohol |
| 11 | **Even a heavy drinker can be successful in getting sober** | Drinking while eating to accompany the meal | Participate in cultural activities instead of drinking alcohol |
| 12 | **Successfully cutting down on alcohol makes you feel better physically** | Feeling like tasting alcohol | Participate in associative activities instead of drinking alcohol |
| 13 | **Successfully cutting down on alcohol makes you feel better mentally** | **Feel like numbing my emotions** | Do a breathing exercise instead of drinking alcohol |
| 14 | **Successfully cutting down on alcohol consumption allows you to free yourself** | Feeling like sleeping | **Making sure not to have alcohol in the house to avoid being tempted to drink** |
| 15 | **Successfully reducing alcohol consumption allows you to make other plans** | Feel like being free | Staying home alone so as not to be tempted to drink |
| 16 | **Alcoholism is a treatable disease** | Feeling like being less shy* | Call someone close to me when I feel like drinking alcohol |
| 17 | **You can learn to manage your drinking, even if you've already had more than you intended** | Feeling fragile | Participate in discussion groups to talk about my drinking |
| 18 | **Successful alcohol management is not a matter of blaming yourself*** | Feeling on edge | **Learn about the effects and consequences of alcohol** |
| 19 | Relapses are an opportunity to learn to manage your drinking* | Feeling sad | **Tell myself that I will not be well the next day** |
| 20 | **People who focus on small changes learn from their failures to manage their drinking*** | Feeling depressed* | Tell myself that I will look bad the next day |
| 21 | **People who focus on small changes develop new skills to manage their drinking*** | **Feeling joyous*** | Tell myself that I won't be able to do sports the next day |
| 22 | **With effort, it is possible to reduce alcohol use*** | Because things are going wrong | **Avoiding situations that make me want to drink alcohol*** |
| 23 | **With the right strategies, it is possible to cut down on drinking*** | Feeling like rewarding myself | **Thinking about consulting health professionals to talk about my alcohol consumption** |
| 24 | It is possible to train yourself to reduce your drinking | **Celebrating a positive event (wedding, birth, promotion,...)** | **Think about serious health problems caused by alcohol** |
| 25 | **Having periods of abstinence from alcohol strengthens the will to manage your drinking** | Feeling marginalized from those who drink | **Think about serious mental health problems caused by alcohol** |
| 26 | Reducing alcohol consumption strengthens the ability to resist the temptation to drink | **In a place where alcohol is easily available (store, supermarket)** | Take medication instead of drinking alcohol |
| 27 | **Abstinence from alcohol strengthens the body's and mind's ability to resist the temptation to drink** | **Being at a time of day when I usually drink** | **Surround myself with people who respect my right not to drink alcohol*** |
| 28 | | Being at work from home | **Think about how drinking hurts those close to me*** |

*(Continued)*

**Table 5.** (Continued)

| N° | Malleable arguments | Critical situations | Appropriate responses |
|---|---|---|---|
| 29 | | **Being offered a drink*** | **Remember the benefits of not drinking alcohol*** |
| 30 | | Being bored | Remember the warnings about the health risks of drinking alcohol |
| 31 | | Being excited* | Tell myself that if I try hard enough you can avoid drinking alcohol* |
| 32 | | Being with people who drink a lot* | Think about who I would be if I controlled my drinking* |
| 33 | | **Things don't go your way*** | Relax |
| 34 | | Being pressured to keep up with others' drinking | Pay attention to alcohol prevention signs in public places* |
| 35 | | **Being with friends** | **Remember the intensity of my feelings when I hurt my loved ones*** |
| 36 | | Being encouraged to drink by others* | **Tell myself that I can choose to change or not change my drinking*** |
| 37 | | Wanting to get over shyness* | **Remember that I feel more competent when I choose not to drink alcohol*** |
| 38 | | **Feeling anxious** | Choose an alcoholic beverage that I don't like |
| 39 | | Feeling angry* | Leave my glass full so I don't refill it |
| 40 | | Things are going really well* | Leave the party or place where I drink alcohol |
| 41 | | Feeling anxious before sex | Pay attention to people who say I shouldn't drink |
| 42 | | Feeling anxious before public speaking | Not talk about alcohol |
| 43 | | Being around alcohol games | Carry a breathalyzer with me |
| 44 | | Feeling hurt | **Tell myself that I have already had a lot to drink** |
| 45 | | Being with someone I like* | **Be the designated driver (Sam)** |
| 46 | | Being nervous about being with other people* | **Say that I have already consumed more than the authorized limit to drive** |
| 47 | | Wanting to be like everyone else | **Say I am taking medication that is incompatible with alcohol** |
| 48 | | Being at a work party | |
| 49 | | Needing to be brave | |
| 50 | | Wanting to be creative | |
| 51 | | **Wanting to forget your problems** | |
| 52 | | Needing to be comforted | |
| 53 | | Not feeling well | |
| 54 | | Feeling lonely | |
| 55 | | Being happy | |
| 56 | | Having problems at work | |
| 57 | | **Feeling like continuing to drink alcohol** | |
| 58 | | Having a body craving for alcohol | |
| 59 | | Feeling tired | |
| 60 | | Being told I shouldn't drink | |
| 61 | | Seeing a bottle of alcohol | |
| 62 | | Having family problems | |

alcohol consumption, a two-tailed between-subject Student's test on groups (clinical vs. general) was performed, $t(95) = -2.44$, $p = .017$, $d = -.51$. The clinical population had a higher growth mindset score ($M = 5.2$, $SD = 0.7$) than the general population ($M = 4.8$, $SD = 0.8$). Correlations revealed that the AUDIT and mindset scores were not correlated, $r = .14$, $p = .16$.

Taken together, these results indicated that the severity of hazardous alcohol consumption differed according to groups, the clinical population (i.e., individuals having undergone medical care specifically for alcohol use disorder) consuming more than the general population. Similarly, differences in mindset was observed between populations, clinical participants having a higher growth mindset (i.e., believing they can change their alcohol consumption) than the general population. However, mindset of alcohol consumption was not directly linked to the intensity of alcohol consumption, meaning that individuals with different levels of consumption (e.g., heavy or light drinker) could hold either a fixed or a growth mindset of alcohol consumption.

**3.4.2. Arguments for the malleability of alcohol consumption.** Table 6 shows the ranking of each mindset argument for each group (clinical vs. general population), according to the aggregated score of agreement and convincingness. The 10 best ranked arguments from the general and the clinical groups were included in the GMII intervention. If an argument was among the 10 bests ranked in both groups, the next best ranked argument was added for inclusion (alternating picks from the clinical and general groups), until 20 arguments in total were selected. Arguments 10 and 22 (see Table 5), ranked 21 and 22, were also added in the final selection because of their rhetorical complementarity with the included arguments 11 and 23. Argument were coupled in pairs in the intervention ("With efforts **and** the right strategies, it is always possible to reduce alcohol use"; "Even a heavy drinker can cut down on their drinking **or** get sober"). Among the selected arguments, individual scores of agreement and convincingness were all significantly positively correlated for the general and clinical populations ($ps <$ .001). The 22 selected arguments are N˚ 12, 13, 2, 14, 16, 15, 11, 1, 9, 23, 25, 27, 6, 8, 21, 18, 3, 4, 20, 17, 10, 22.

**3.4.3. Critical situations.** Table 6 shows the ranking of each situation for each group, according to the aggregated score of frequency of encounter and susceptibility to consume. The same selection method as for the malleable arguments was used until a total of 20 situations was reached. Among the selected situations, individual scores of frequency of encounter and susceptibility to consume were all significantly positively correlated ($ps <$ .05) except situation 21 in the general group ($r = .17$, $p = .21$) and situations 6 ($r = .16$, $p = .44$), 26 ($r = .32$, $p =$ .12), 9 ($r = .38$, $p = .07$), and 33 ($r = .36$, $p = .08$) in the clinical group. Because the frequency and susceptibility to consume scores remained high ($Ms > 3.38$, scale median point = 3.5), these situations were retained. The 20 selected situations are N˚ 13, 35, 57, 3, 38, 5, 51, 24, 6, 4, 18, 2, 27, 8, 26, 29, 9, 21, 33, 1.

**3.4.4. Appropriate responses.** Table 6 shows the ranking of each response for each group, according to the aggregated score of effectiveness and self-efficacy. The same selection method as for arguments and situations was used. For the clinical group, scores were significantly positively correlated ($ps <$ .05) except for responses 23 ($r = .07$, $p = .74$) and 5 ($r = .32$, $p = .12$). For both these situations, frequency and susceptibility to consume scores remained high and were retained (above the scale median point = 3.5). The 20 selected responses are N˚ 23, 45, 7, 27, 14, 29, 25, 24, 4, 37, 46, 5, 19, 28, 44, 18, 36, 22, 35, 47.

## 3.5. Discussion

Selected arguments, situations and responses for the final GMII intervention reflected and conserved the diversity of data collected in study 1, since each study 1's subtheme was represented in the present selection in the form of arguments, situations and responses.

Selected material can also be compared to the current literature on GM and II interventions. If no GM intervention targeting alcohol consumption existed to our knowledge at the time of the study [22], there was one interventional GM study targeting smoking cessation

**Table 6. Ranking of each argument, situation and response for each group (clinical vs. general population).**

| Rank | General group (N = 60) Argument N° | Mean | SD | Clinical group (N = 37) Argument N° | Mean | SD | General group (N = 56) Situation N° | Mean | SD | Clinical group (N = 24) Situation N° | Mean | SD | General group (N = 55) Response N° | Mean | SD | Clinical group (N = 24) Response N° | Mean | SD |
|---|---|---|---|---|---|---|---|---|---|---|---|---|---|---|---|---|---|---|
| 1 | 12 | 5.28 | 0.87 | 12 | 5.51 | 0.76 | 35 | 4.00 | 1.29 | 13 | 4.67 | 1.32 | 45 | 4.80 | 1.62 | 23 | 5.44 | 0.68 |
| 2 | 13 | 5.18 | 0.90 | 13 | 5.46 | 0.79 | 3 | 3.96 | 1.30 | 57 | 4.65 | 1.36 | 27 | 4.77 | 1.45 | 7 | 5.02 | 1.10 |
| 3 | 2 | 5.09 | 0.88 | 2 | 5.43 | 0.79 | 5 | 3.94 | 1.39 | 38 | 4.58 | 1.42 | 14 | 4.74 | 1.31 | 27 | 4.98 | 1.19 |
| 4 | 16 | 5.09 | 0.82 | 14 | 5.36 | 0.83 | 24 | 3.86 | 1.22 | 51 | 4.52 | 1.30 | 7 | 4.65 | 1.53 | 29 | 4.96 | 1.14 |
| 5 | 15 | 4.98 | 1.03 | 15 | 5.23 | 0.88 | 4 | 3.82 | 1.11 | 6 | 4.40 | 1.24 | 29 | 4.65 | 1.30 | 25 | 4.90 | 1.07 |
| 6 | 1 | 4.98 | 0.89 | 11 | 5.11 | 1.29 | 2 | 3.75 | 1.24 | 18 | 4.38 | 1.25 | 4 | 4.65 | 1.37 | 24 | 4.90 | 0.88 |
| 7 | 9 | 4.98 | 0.97 | 16 | 5.07 | 1.37 | 8 | 3.74 | 1.22 | 27 | 4.35 | 1.65 | 46 | 4.65 | 1.61 | 37 | 4.83 | 1.16 |
| 8 | 23 | 4.97 | 0.76 | 25 | 4.96 | 1.22 | 29 | 3.62 | 1.18 | 26 | 4.33 | 1.20 | 19 | 4.57 | 1.32 | 5 | 4.69 | 1.39 |
| 9 | 14 | 4.88 | 1.02 | 27 | 4.95 | 1.13 | 9 | 3.52 | 1.36 | 8 | 4.33 | 1.46 | 44 | 4.51 | 1.24 | 28 | 4.69 | 1.32 |
| 10 | 11 | 4.81 | 0.97 | 9 | 4.93 | 1.07 | 21 | 3.46 | 1.06 | 9 | 4.29 | 1.16 | 36 | 4.48 | 1.30 | 18 | 4.67 | 1.41 |
| 11 | 8 | 4.76 | 1.13 | 6 | 4.91 | 1.28 | 1 | 3.38 | 1.08 | 33 | 4.29 | 1.22 | 28 | 4.44 | 1.49 | 14 | 4.60 | 1.46 |
| 12 | 18 | 4.73 | 0.90 | 23 | 4.91 | 1.04 | 55 | 3.38 | 1.04 | 20 | 4.27 | 1.25 | 37 | 4.44 | 1.45 | 36 | 4.58 | 1.30 |
| 13 | 6 | 4.72 | 0.96 | 21 | 4.88 | 0.90 | 7 | 3.34 | 1.35 | 23 | 4.27 | 1.36 | 24 | 4.41 | 1.45 | 22 | 4.48 | 1.31 |
| 14 | 4 | 4.72 | 1.00 | 3 | 4.88 | 1.16 | 45 | 3.28 | 1.14 | 61 | 4.25 | 1.08 | 22 | 4.35 | 1.32 | 35 | 4.40 | 1.56 |
| 15 | 17 | 4.67 | 0.84 | 20 | 4.84 | 0.93 | 32 | 3.28 | 1.35 | 19 | 4.23 | 1.28 | 47 | 4.34 | 1.67 | 19 | 4.33 | 1.46 |
| 16 | 10 | 4.65 | 0.94 | 18 | 4.82 | 1.19 | 40 | 3.23 | 1.05 | 35 | 4.21 | 1.35 | 11 | 4.31 | 1.49 | 32 | 4.33 | 1.51 |
| 17 | 22 | 4.65 | 0.94 | 4 | 4.80 | 1.18 | 26 | 3.21 | 1.24 | 53 | 4.19 | 1.33 | 25 | 4.26 | 1.58 | 33 | 4.31 | 1.21 |
| 18 | 21 | 4.61 | 0.79 | 1 | 4.73 | 1.25 | 23 | 3.17 | 1.37 | 54 | 4.17 | 1.52 | 5 | 4.25 | 1.60 | 17 | 4.29 | 1.39 |
| 19 | 25 | 4.60 | 0.97 | 5 | 4.73 | 1.01 | 11 | 3.12 | 1.24 | 5 | 4.10 | 1.38 | 35 | 4.25 | 1.52 | 30 | 4.29 | 1.26 |
| 20 | 24 | 4.58 | 0.91 | 10 | 4.73 | 1.26 | 48 | 3.11 | 1.51 | 39 | 4.08 | 1.45 | 31 | 4.18 | 1.36 | 11 | 4.27 | 1.53 |
| 21 | 3 | 4.54 | 1.12 | 7 | 4.62 | 1.17 | 15 | 3.04 | 1.20 | 52 | 4.08 | 1.03 | 8 | 4.17 | 1.47 | 47 | 4.27 | 1.54 |
| 22 | 20 | 4.53 | 0.86 | 19 | 4.61 | 1.13 | 61 | 3.02 | 1.26 | 10 | 4.06 | 1.57 | 18 | 4.17 | 1.41 | 20 | 4.27 | 1.39 |
| 23 | 27 | 4.53 | 1.07 | 8 | 4.50 | 1.43 | 6 | 2.94 | 1.18 | 24 | 4.02 | 1.28 | 30 | 4.17 | 1.38 | 31 | 4.25 | 1.36 |
| 24 | 7 | 4.51 | 1.21 | 22 | 4.46 | 1.23 | 52 | 2.81 | 1.26 | 3 | 3.98 | 1.47 | 32 | 4.15 | 1.53 | 10 | 4.23 | 1.58 |
| 25 | 5 | 4.46 | 0.94 | 24 | 4.39 | 1.24 | 59 | 2.80 | 1.19 | 22 | 3.98 | 1.23 | 39 | 4.14 | 1.70 | 8 | 4.19 | 1.55 |
| 26 | 26 | 4.38 | 1.01 | 26 | 4.20 | 1.28 | 51 | 2.79 | 1.33 | 7 | 3.92 | 1.84 | 2 | 4.13 | 1.40 | 26 | 4.15 | 1.70 |
| 27 | 19 | 4.09 | 1.04 | 17 | 4.19 | 1.38 | 57 | 2.79 | 1.61 | 29 | 3.90 | 1.37 | 9 | 4.11 | 1.57 | 13 | 4.06 | 1.64 |
| 28 | | | | | | | 31 | 2.77 | 1.14 | 59 | 3.88 | 1.03 | 3 | 4.10 | 1.43 | 12 | 3.96 | 1.42 |
| 29 | | | | | | | 34 | 2.74 | 1.40 | 17 | 3.83 | 1.50 | 33 | 4.05 | 1.45 | 40 | 3.96 | 1.59 |
| 30 | | | | | | | 19 | 2.73 | 1.28 | 32 | 3.77 | 1.52 | 12 | 3.97 | 1.56 | 45 | 3.83 | 1.90 |
| 31 | | | | | | | 38 | 2.72 | 1.28 | 4 | 3.75 | 1.14 | 20 | 3.96 | 1.56 | 3 | 3.71 | 1.46 |
| 32 | | | | | | | 20 | 2.67 | 1.30 | 44 | 3.71 | 1.55 | 42 | 3.86 | 1.48 | 1 | 3.63 | 1.62 |
| 33 | | | | | | | 33 | 2.67 | 1.09 | 30 | 3.71 | 1.47 | 23 | 3.81 | 1.53 | 9 | 3.54 | 1.62 |
| 34 | | | | | | | 18 | 2.64 | 1.09 | 2 | 3.65 | 1.37 | 21 | 3.71 | 1.69 | 15 | 3.46 | 1.67 |
| 35 | | | | | | | 36 | 2.63 | 1.36 | 15 | 3.60 | 1.56 | 40 | 3.69 | 1.59 | 39 | 3.44 | 1.53 |
| 36 | | | | | | | 49 | 2.63 | 1.31 | 21 | 3.56 | 1.36 | 43 | 3.67 | 1.63 | 4 | 3.44 | 1.26 |
| 37 | | | | | | | 10 | 2.60 | 1.57 | 58 | 3.54 | 1.85 | 1 | 3.65 | 1.57 | 44 | 3.40 | 1.53 |
| 38 | | | | | | | 14 | 2.58 | 0.99 | 16 | 3.54 | 1.81 | 15 | 3.65 | 1.44 | 34 | 3.33 | 1.72 |
| 39 | | | | | | | 22 | 2.57 | 1.22 | 49 | 3.52 | 1.65 | 16 | 3.61 | 1.41 | 41 | 3.33 | 1.87 |
| 40 | | | | | | | 54 | 2.55 | 1.24 | 31 | 3.50 | 1.53 | 17 | 3.50 | 1.52 | 2 | 3.21 | 1.49 |
| 41 | | | | | | | 53 | 2.53 | 1.25 | 45 | 3.50 | 1.14 | 41 | 3.48 | 1.59 | 46 | 3.17 | 1.75 |
| 42 | | | | | | | 56 | 2.50 | 1.28 | 62 | 3.50 | 1.67 | 10 | 3.45 | 1.68 | 16 | 3.17 | 1.79 |
| 43 | | | | | | | 13 | 2.43 | 1.45 | 14 | 3.48 | 1.50 | 6 | 3.43 | 1.71 | 42 | 3.10 | 1.55 |
| 44 | | | | | | | 50 | 2.43 | 1.12 | 1 | 3.46 | 1.75 | 13 | 3.41 | 1.40 | 38 | 2.96 | 1.57 |
| 45 | | | | | | | 44 | 2.42 | 1.17 | 11 | 3.46 | 1.69 | 34 | 3.37 | 1.69 | 21 | 2.85 | 1.60 |

*(Continued)*

**Table 6.** (Continued)

| Rank | General group (N = 60) | | | Clinical group (N = 37) | | | General group (N = 56) | | | Clinical group (N = 24) | | | General group (N = 55) | | | Clinical group (N = 24) | | |
|---|---|---|---|---|---|---|---|---|---|---|---|---|---|---|---|---|---|---|
| | Argument N° | Mean | SD | Argument N° | Mean | SD | Situation N° | Mean | SD | Situation N° | Mean | SD | Response N° | Mean | SD | Response N° | Mean | SD |
| 46 | | | | | | | 39 | 2.41 | 1.16 | 12 | 3.42 | 1.68 | 38 | 2.78 | 1.59 | 6 | 2.65 | 1.54 |
| 47 | | | | | | | 62 | 2.41 | 1.18 | 37 | 3.38 | 1.64 | 26 | 2.52 | 1.57 | 43 | 2.60 | 1.65 |
| 48 | | | | | | | 17 | 2.39 | 1.32 | 40 | 3.35 | 1.32 | | | | | | |
| 49 | | | | | | | 46 | 2.39 | 1.30 | 56 | 3.29 | 1.83 | | | | | | |
| 50 | | | | | | | 16 | 2.38 | 1.46 | 60 | 3.29 | 1.66 | | | | | | |
| 51 | | | | | | | 12 | 2.38 | 1.51 | 55 | 3.25 | 1.29 | | | | | | |
| 52 | | | | | | | 30 | 2.36 | 1.19 | 34 | 3.10 | 1.70 | | | | | | |
| 53 | | | | | | | 42 | 2.33 | 1.23 | 46 | 3.10 | 1.73 | | | | | | |
| 54 | | | | | | | 27 | 2.31 | 1.44 | 47 | 2.92 | 1.61 | | | | | | |
| 55 | | | | | | | 37 | 2.27 | 1.33 | 36 | 2.90 | 1.49 | | | | | | |
| 56 | | | | | | | 47 | 2.25 | 1.20 | 50 | 2.71 | 1.34 | | | | | | |
| 57 | | | | | | | 43 | 2.24 | 1.51 | 25 | 2.60 | 1.43 | | | | | | |
| 58 | | | | | | | 60 | 2.05 | 1.22 | 42 | 2.60 | 1.35 | | | | | | |
| 59 | | | | | | | 25 | 1.88 | 0.90 | 48 | 2.42 | 1.32 | | | | | | |
| 60 | | | | | | | 41 | 1.82 | 1.17 | 41 | 2.25 | 1.44 | | | | | | |
| 61 | | | | | | | 28 | 1.75 | 0.88 | 28 | 2.02 | 1.56 | | | | | | |
| 62 | | | | | | | 58 | 1.60 | 1.15 | 43 | 1.83 | 1.45 | | | | | | |

[30]. Authors extracted 6 fixed beliefs about nicotine addiction through literature search. The beliefs were that addiction is permanent because: (1) it is genetic; (2) some people have an addictive personality; (3) it irreversibly changes the brain; (4) withdrawal symptoms may persist after cessation; (5) people can feel like smoking even years after quitting; (6) failure to quit smoking is indicative of a permanent habit. Compared to it, the present material included arguments related to the malleability of (2) personality (N°3), (4) withdrawal symptoms (N° 12–13) [63], (5) cravings (N°25, 27) [61, 70], and (6) habits (N°15). In another study manipulating GM of drug and alcohol addiction [23], authors created a "compensatory-growth" GM message. The message contained arguments defending that it is possible to learn to manage one's addiction with efforts and the right strategies, that setbacks are an opportunity to grow and develop new skills, and that it is a matter of not blaming oneself. Based on this material, 5 arguments were selected in the present study (N°18, 20, 21, 22, 23) for inclusion in the GMII intervention. The message also mentioned addiction as a manageable disease (N°16) [66], while this argument was also discussed in the focus groups (see study 1). The present material went further by including arguments related to social representations of alcohol (N°6, 8, 9) [65], or that successful recovery from drinking abuse, and becoming a functioning member of society is possible [62] (arguments N°1, 3, 10, 11).

Compared to existing VHS targeting alcohol consumption [44, 47], 3 situations (N°21, 29, 33) and 7 responses (N°22, 27, 28, 29, 35, 36, 37) remained. Selected situations covered: social situations ("being with family" (N°1) [73], friends (N°35) [74], or being isolated (N°6) [75]), habits (i.e., usual moment of the day (N°27) [76]), negative events (N°33, 51) [71] and states (N° 13, 38) [71, 78, 80], and sensation seeking (N°9) [77]. Selected appropriate responses covered BCT strategies known to be effective to improve alcohol outcomes [82], such monitoring strategies (N°44) or social support [81] (N°23, 27). Other known promising BCT for alcohol misuse [83] were included, such as avoidance/reducing exposure to cues for behavior techniques (N°14, 22) and "listing pros and cons" (N°19, 24, 28, 29, 37).

## 4. General discussion

This work aimed to create a psychosocial intervention based on growth mindset and implementation intention (GMII) targeting the reduction of alcohol consumption in the clinical and general populations. To design the intervention, focus groups were first used among these populations. It allowed to collect convincing arguments about the malleability of alcohol consumption (study 1A), as well as critical situations and appropriate responses linked to alcohol consumption (study 1B). Then, the most persuasive arguments and the most appropriate situations and responses were extracted (study 2) to include in the final intervention. Results from study 1 showed that the variety of themes discussed in the groups reflected what can be found in the literature on alcohol consumption. The arguments, situations and responses selected in study 2 maintained and confirmed this diversity. In sum, the GMII intervention, created using both empirical evidence and input from the literature, condensed a variety of arguments, situations and responses known to be relevant to alcohol consumption and management.

### 4.1. Description of the GMII intervention

The first part of the intervention presents the popularized scientific article describing alcohol consumption as malleable, followed by two internalization exercises. The text contains all 22 selected arguments, arranged in a coherent way in five paragraphs. The introduction starts by telling the reader that it is possible to reduce or stop one's alcohol consumption. The article then describes the several benefits one might gain by reducing alcohol consumption. The following paragraph focus on changing social representations about alcohol, such as needing alcohol to party, or having to drink to be included in the group. Then, the text describes how willpower and motivation play a role to reduce consumption. Finally, some advice is given on how to reduce alcohol consumption. In accordance with the literature, the text contains three testimonies from experts [23], and one testimony from a person diagnosed with AUD (acting as a peer testimony [84]). In the first internalization exercise, individuals have to recall and write in a few lines a personal example of a situation or event in which they successfully reduced their alcohol consumption. In the second exercise, they have to write a letter of encouragement to someone with the same level of alcohol consumption as themselves, who could be experiencing difficulties diminishing their consumption. They have to use the arguments presented in the text. These two exercises have been used in GM interventions in the past [4, 19, 20]. The VHS is presented following the GM article. It contains on a single page the 20 selected situations (left column) and the 20 selected responses (right column). Individuals are asked to read the set of situations and the set of solutions, to choose a situation that seems most relevant to them and that they encounter the most frequently, and to connect it with a line to the solution that they find most appropriate to reduce their alcohol consumption. The drawing of a line reflects the conceptualization of Gollwitzer [31] of implementation intentions as links between critical situations and appropriate responses. They are asked to do this exercise three times, and to copy each of the three proposals in the three lines at the bottom of the page. The VHS structure follows tested material from the literature [44, 47]. We chose to set the number of plans to three, as to allow individuals to formulate plans in response to a diversity of contexts, which often act as triggers for alcohol consumption [76], while not asking too much as to not overwhelm them.

In total, the intervention is 5 pages long, and its format allows its utilization in different contexts. For example, as prevention and sensibilization tool, the GMII intervention could be accessible on alcohol prevention websites and centers as a self-administered tool to help reduce consumption, or during certain events such as the Dry January (*Janvier Sobre* in France, gathering a growing number of participants wanting to reduce or stop their alcohol consumption

for one month). For the clinical population, its light and easily administrable format would allow the GMII to be used flexibly with patients, whether it be completed autonomously or with the help of the medical staff as part of therapeutic sessions.

## 4.2. Validation of the GMII intervention

The efficacy of the GMII intervention to reduce alcohol consumption will be tested in a future study on the general population of drinkers and the clinical population of individuals with AUD. Compared to a control condition, we will measure its efficacy on reducing alcohol consumption on short- and long-term (baseline; 1 month; 3 months; 6 months after intervention) and its impact on potential psychological mediators and outcome variables such as mindset of alcohol consumption, motivation, intention and perceived control to reduce consumption, anxiety and depression, and well-being. If proven effective, it would also be possible to use the same design method (i.e., focus groups and questionnaires) to develop a GMII intervention targeting another substance use (e.g., cannabis, cocaine, etc.) or behavior (e.g., problematic gambling).

## 4.3. Limitations

This work is not without limitations. Most notably, we must take note of the reduced number of focus groups (4 focus groups in total, $N = 28$), as well as participants in study 2 ($N = 97$). If the qualitative data collected was rich, it could nonetheless be hypothesized that more themes reflecting diverse beliefs, situations and responses related to alcohol consumption might have emerged with more groups. Cultural aspects of drinking, for example, might influence beliefs and subsequent drinking [64]. If culture was mentioned by participants ("social norms of alcohol" component in group 1A and "culture of alcohol" component in the group 1B), more groups would have allowed further room for cultural differences to be expressed and integrated into the intervention. This could also have allowed to more clearly differentiate between central themes, discussed by a majority of participants, and minor ones, which was here limited. Study 2 was conceived to counter this lack of hierarchical, quantitative ranking. While it allowed to select the best ranked items, the results might have been considered more robust with a larger sample.

Attrition was high, probably due to the length of the two-part questionnaire. Indeed, study 2 was done online and autonomously by participants. Despite the 20€ reward incentive to complete the study, the remote and dense nature of the task might have discouraged engagement. Technical issues preventing access to the questionnaire (e.g., internet connection, email errors) might also have occurred. A potential solution would have been to reduce the questionnaire's length, or to invite participants to do the study in person in a laboratory context. The recruitment platform used for the general population (i.e., the Risc list) presented several advantages such as convenience, reach, and cost-effectiveness. However, it also carried potential limitations and biases, such as a non-representative population (as it is originally an academic platform, a lot of participants are students), and an incentive to participate due to the financial reward, which might have negatively impacted participants' engagement and responses quality. Furthermore, this work's samples contained more females than males. This might not accurately represent the clinical or general populations of drinkers, still predominantly male [7]. Nonetheless, research show a recent increase in women's alcohol consumption, despite sex and gender differences still existing regarding the trajectory of hazardous alcohol consumption, its clinical consequences, and social representations of alcohol [86].

Study 1 showed the diversity of beliefs, in particular the negative beliefs, participants hold about alcohol consumption and dependence. This result contrasted with the quantitative

scores observed in study 2. Indeed, mindset scores remained high for both groups. However, the measure effectively differentiated groups, the clinical population having a higher growth mindset than the general population. This higher score for the clinical group might mean that individuals diagnosed with AUD and engaged in a therapeutic process were rightfully convinced that they can change and improve their alcohol consumption. Conversely, it could also mean that they lacked insight about their alcohol consumption, and failed to see the problem they had with alcohol. This would reflect a common discourse among patients with AUD, namely that they can stop drinking whenever they want and that they do not have an alcohol problem.

The constructed intervention focused on changing two determinants of alcohol consumption: beliefs about alcohol, and behavioral responses of individuals in relation to their consumption. Alcohol abuse is a complex phenomenon, with physiological (e.g., genetics, heritability, ethnicity) as well as socio-cultural (e.g., past traumas, education, socio-economic status) factors at play. Consequently, it is important to note that the intervention did not aim to address all facets of alcohol consumption, but only these two determinants. This focus on beliefs and behavioral responses might help inform potential obstacles to the success of existing prevention programs and therapy. Relatedly, creating an intervention targeting both the general and clinical populations might have "diluted" the data, meaning that it could lack efficacy for the clinical population, presenting specific needs, because it was also adapted to the general population. Finally, failure to reach one's desired objectives (abstinence from alcohol or reduced drinking) following the intervention might cause a blowback and lead to further discouragement and guilt from people with AUD. Without tests of the efficacy of the intervention in the general and clinical populations, the tool should be used in conjunction with other proven methods such as support groups, therapy, or medication. All these limitations will be the subject of special attention in the test of the efficacy of the intervention.

### 4.4. Conclusion

This work aimed at creating a psychosocial intervention designed to help reduce alcohol consumption for both the general population of drinkers and the clinical population of individual suffering from AUD. Using qualitative (focus groups) and quantitative (questionnaires) research methods, we created a new type of psychosocial intervention, combining fitting behavior change theories and strategies (growth mindset and implementation intention), and aligned with the scientific knowledge on alcohol consumption. The resulting GMII intervention has an easy-to-use and easily administrable format, and is to be tested among the general and clinical populations in a future longitudinal study.

## Supporting information

**S1 Checklist. STROBE statement—Checklist of items that should be included in reports of *cross-sectional studies*.**
(DOC)

## Acknowledgments

We would like to thank the staff of Sainte-Anne Hospital's *CSAPA* for their helpful contribution to the recruitment of the clinical population. We would also like to thank Laurence Cottet, president of the association *H3D / France Janvier Sobre* [H3D / Dry January France] for her assistance in the recruitment of participants and the creation of the intervention.

## Author Contributions

**Conceptualization:** Sacha Parada, Jean-François Verlhiac, Eve Legrand.

**Formal analysis:** Bérengère Rubio.

**Funding acquisition:** Sacha Parada, Jean-François Verlhiac, Eve Legrand.

**Investigation:** Sacha Parada.

**Methodology:** Sacha Parada, Bérengère Rubio, Jean-François Verlhiac.

**Project administration:** Eve Legrand.

**Resources:** Xavier Laqueille.

**Supervision:** Jean-François Verlhiac, Eve Legrand.

**Validation:** Bérengère Rubio, Elsa Taschini, Xavier Laqueille, Malika El Youbi, Pierre Paris, Bernard Angerville, Alain Dervaux, Jean-François Verlhiac, Eve Legrand.

**Writing – original draft:** Sacha Parada.

**Writing – review & editing:** Sacha Parada, Bérengère Rubio, Elsa Taschini, Xavier Laqueille, Malika El Youbi, Pierre Paris, Bernard Angerville, Alain Dervaux, Jean-François Verlhiac, Eve Legrand.

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
