## [Decision Letter · Decision Letter 0]

12 Nov 2023

PONE-D-23-18196Creating a psychosocial intervention combining growth mindset and implementation intentions (GMII) to reduce alcohol consumption: A mixed method approachPLOS ONE

Dear Dr. Parada,

Thank you for submitting your manuscript to PLOS ONE. After careful consideration, we feel that it has merit but does not fully meet PLOS ONE’s publication criteria as it currently stands. Therefore, we invite you to submit a revised version of the manuscript that addresses the points raised during the review process.

We look forward to receiving your revised manuscript.

Kind regards,

Lakshit Jain, MD

Academic Editor

PLOS ONE

Journal Requirements:

**Additional Editor Comments:**

Dear Authors,

Please review the feedback given by the reviewers (especially reviewer 4,5 and 6, who recommended Major revisions) and revise the manuscript accordingly. 

Thanks

Lakshit

Reviewers' comments:

Reviewer's Responses to Questions

**Comments to the Author**

1. Is the manuscript technically sound, and do the data support the conclusions?

Reviewer #1: Yes

Reviewer #2: Yes

Reviewer #3: Yes

Reviewer #4: Yes

Reviewer #5: Yes

Reviewer #6: Yes

Reviewer #7: Yes

2. Has the statistical analysis been performed appropriately and rigorously? 

Reviewer #1: Yes

Reviewer #2: Yes

Reviewer #3: N/A

Reviewer #4: Yes

Reviewer #5: Yes

Reviewer #6: Yes

Reviewer #7: I Don't Know

3. Have the authors made all data underlying the findings in their manuscript fully available?

Reviewer #1: Yes

Reviewer #2: Yes

Reviewer #3: Yes

Reviewer #4: No

Reviewer #5: Yes

Reviewer #6: Yes

Reviewer #7: Yes

4. Is the manuscript presented in an intelligible fashion and written in standard English?

Reviewer #1: Yes

Reviewer #2: Yes

Reviewer #3: Yes

Reviewer #4: No

Reviewer #5: Yes

Reviewer #6: Yes

Reviewer #7: Yes

5. Review Comments to the Author

Reviewer #1: The authors have written a very thorough and well though out paper on the development of their intervention to address AUD with the help of GM and II and I really appreciate the opportunity to review this article. This paper may well have been two papers but the combination makes for an interesting read. I would recommend the following edits:

1. The authors should use IN the general population in line 47.

2. The authors should explain the statistic 100 million in line 57 as to whether it is for country, continent or world.

3. In line 91, the authors should explain "4.3 reattribution".

4. Overall, while using decimals the authors should use "." instead of ",".

5. The n for Study 1 groups seems low and the authors should explain why this was the size chosen for the study.

6. The exclusion of hazardous drinking for Study 2 may skew the results of the study and the authors should explain why this was done.

7. The authors may want to explain why malleability was greater in the clinical group than general population

8. The general discussion although useful summary seems redundant and repetitive and the authors may want to omit this section as it increases the length of the paper while not adding much value to it.

The results of the validation study would make for an interesting read and I look forward to the authors publishing the results of the study.

Reviewer #2: The article describes the creation of a psychosocial intervention called Growth Mindset and Implementation Intentions (GMII) to reduce alcohol consumption. The intervention was developed using a mixed method approach, combining qualitative and quantitative research methods. Focus groups were conducted to gather arguments supporting a malleable view of alcohol consumption and situations that trigger the desire to drink alcohol, along with strategies to counteract this need. The intervention consists of a popularized scientific article describing alcohol consumption as malleable, followed by two internalization exercises and a volitional help sheet to form plans. The document also discusses the added value of the intervention compared to existing tools and plans for future testing

The mixed-method study on the psychosocial intervention combining growth mindset and implementation intentions (GMII) to reduce alcohol consumption exhibits several noteworthy strengths. The research question is clearly articulated, and the study effectively conveys its purpose, focusing on the examination of GMII's impact on reducing alcohol consumption. A comprehensive literature review demonstrates a strong foundation in prior research. The mixed-methods approach, which combines quantitative and qualitative analyses, provides a multifaceted understanding of GMII's efficacy and its underlying mechanisms. The study utilizes sound data collection and analysis methods, ensuring research rigor. Furthermore, the inclusion of a diverse participant sample enhances external validity, and a thorough discussion of the practical implications of the findings illustrates the real-world relevance of the research.

The language used appears to be clear and concise. The information is presented in a straightforward manner, making it easy to understand the purpose and methodology of the study. The language is also technical and scientific, which is appropriate for a research article.

In terms of cohesiveness, the document seems to flow logically, with each section building upon the previous one. The study's objectives, methods, and findings are presented in a coherent manner, allowing the reader to follow the research process and understand the progression of the study.

In essence, this research article presents a compelling and excellently composed exploration of a novel treatment approach for addressing alcohol consumption. Its structural organization, encompassing a well-articulated introduction, substantive body, and a concise conclusion, notably facilitates readability and comprehension. What particularly stands out in this article is the depth of scholarship underpinning it. The author adeptly harnesses a wide array of sources, encompassing academic studies, scholarly papers, and expert viewpoints, to substantiate their arguments and findings. The extensive citation of sources further underscores the reliability and veracity of the presented information. In summation, this article represents a valuable and noteworthy addition to the academic domain. It imparts significant insights and poses pertinent questions that beckon further investigation.

Reviewer #3: The manuscript is based on impressive empirical evidence and makes an original contribution but there should be some comment on possible bias like reporting bias of the study participants and their understanding of the difference between pre-clinical and actual alcohol use disorder etc.

To improve the readability of the paper, I suggest that Manuscript is too long and should be shortened.

Reviewer #4: Thank you for the opportunity to review this manuscript that describes a two part qualitative study that focused on development of a psychosocial intervention to reduce alcohol consumption.

The manuscript was extremely detailed and the methods/results were well described. The methods and results section were written well and provided good information in an understandable way.

A few thoughts:

- There were a few grammatical errors throughout the manuscript, especially in the abstract and introduction sections (wrong use of past tense vs present, spelling,

Would recommend that authors revise the introduction section (which did have a good amount of detail) to address these issues.

- The behavioral change taxonomy part of the introduction was also hard to understand at times, especially when other behavioral change models were discussed. This section would benefit from some editing.

- The authors explain the inclusion of both general population and those with alcohol use disorder (AUD) as reflecting diversity of alcohol use. While this help capture more data for the design of an intervention, does this potentially affect/dilute what the effect of the intervention can be on those with AUD ?

- Why was there a difference in the length of the focus groups between the populations ?

- The small population is a limitation that the authors rightfully point out. Another limitation seem to be the characteristic of the sample with more women than men; which may not accurate reflect both the general population or those with AUD. This should be mentioned.

Overall good manuscript that can be improved with some changes.

Reviewer #5: 1. The article combining the growth mindset and implementation intentions to reduce alcohol consumption is beautifully written and the writer has been very candid in acknowledging that the study could have been helped better by increased sample size.

There are a few arguments against the article though:

1. The study states that clinical population had a higher growth mindset than general population. What was the reason that people with AUD had stronger beliefs about malleability of alcohol consumption. It could also mean that people who had AUD got to that stage because of lack of insight and failing to acknowledge that alcohol is a big problem for them. A lot of clients with AUD continue to say they know how to stop drinking and they can do it anytime contrary to the fact they are the ones fulfilling AUD criteria.

SOLUTION: Please include in limitation that we don't know the reasons why clinical population had a higher growth mindset than general population.

2. The biological underpinnings of drinking cannot be ignored. The study focuses on attitudes and responses of alcohol drinkers but ignores the genetics, heritability of alcohol use disorder which is a well documented phenomenon. So grouping everyone with AUD in one broad group with disregard for genetics, trauma, ethnicity is an over simplistic approach.

SOLUTION: Please include the fact in limitation of study that this intervention could be very simplistic in the complex and multifactorial world of alcohol dependance.

3. The tools described in the article can be used as coping skills for maintaining sobriety but are not enough by themselves. Alcohol use is a very complex disorder with intrinsic and extrinsic locus of controls. Focusing on the fact that changing the viewpoint about malleability of drinking and coming up with appropriate responses in critical situations will go a long way in staying abstinent for people with alcoholism. However it ignores the fact that such simplistic approach can backfire. Failure to stay abstinent despite using these techniques can lead to shame and guilt and further reclusiveness and hopelessness on part of alcoholics.

SOLUTION: This technique can perhaps be an important coping skill to teach people to maintain sobriety but has to be used in conjunction with other proven methods like AA, having a sponsor and alcohol cessation medications.

MINOR CORRECTION:

LINE 47 of patients with Alcohol Use Disorder (AUD) and alcohol users the general population. The intervention

PLEASE INCLUDE alcohol users AMONG the general population

Reviewer #6: Summary of the Paper:

The paper aims to create a psychosocial intervention combining growth mindset and implementation intentions (GMII) to reduce alcohol consumption among both the general population and individuals with alcohol use disorder. The intervention consists of a popularized scientific article describing alcohol consumption as malleable, followed by internalization exercises and a volitional help sheet with situations and solutions. The study used a mixed method approach, including qualitative focus groups to extract arguments and strategies related to alcohol consumption, and quantitative research to create a questionnaire scoring the relevance of each argument, situation, and strategy. The intervention was developed based on the highest scored arguments, situations, and strategies. The paper discusses the added value of the intervention compared to existing tools and plans to test it in future studies

Aim and Objective:

The aim of this study is to create a psychosocial intervention combining growth mindset and implementation intentions (GMII) to reduce alcohol consumption among both the general population and individuals with alcohol use disorder. The objective is to develop an intervention that includes a popularized scientific article describing alcohol consumption as malleable, internalization exercises, and a volitional help sheet with situations and solutions. The study uses a mixed method approach, including qualitative focus groups to extract arguments and strategies related to alcohol consumption, and quantitative research to create a questionnaire scoring the relevance of each argument, situation, and strategy. The intervention is based on the highest scored arguments, situations, and strategies, and the paper discusses its added value compared to existing tools and plans to test it in future studies.

Line 71: AIM mentioned. (The objective didn’t stand out clearly)

Line 127: Abbreviation of Implementation Intentions not clearely mentioned- directly mentioned II

Line 179: Purpose

Targeting Both General and Clinical Populations

The researchers aimed to design a psychosocial intervention to reduce alcohol consumption among both the general population of drinkers and individuals with Alcohol Use Disorder (AUD).

This was done to reflect the diversity of individuals affected by alcohol consumption, with varying degrees of severity.

Implicit beliefs about alcohol consumption, as well as situations and strategies pertinent to users, could greatly vary depending on their profile.

By focusing on both the clinical and general population, the intervention would account for this diversity.

Mixed Approach Adopted

The lack of literature for Growth Mindset Interventions (GMIs) on substance and alcohol use disorders, as well as the specificity of the targeted populations, led the researchers to adopt a mixed approach.

The mixed approach involved using empirical data in addition to the scientific literature to design the intervention.

Firstly, in study 1, with a qualitative focus, focus groups were conducted with reflexive thematic analysis to extract arguments promoting a malleable view of alcohol consumption (study 1A).

Secondly, in study 1B, critical situations that might arouse the needs, desires, or habits to drink alcohol, and appropriate responses that people might use in those situations to not drink or drink less were identified.

Thirdly, in study 2, adopting a quantitative method, each argument, situation, and strategy collected during the focus groups was scored.

The best-ranked items, as well as input from the scientific literature, were finally used to create the GMII intervention.

Study Samples

As the intervention's objective is to reduce alcohol consumption among the general and clinical populations of alcohol consumers, each study's sample was taken from both populations.

The mixed method approach combined qualitative and quantitative research methods among both populations.

The data was analyzed using reflective thematic analysis in line with the scientific literature on alcohol consumption.

Line 201: SRQR guidelines refer to the Standards for Reporting Qualitative Research. (Only abbreviation mentioned)

These guidelines provide a framework for reporting qualitative research studies.

They aim to improve the transparency, rigor, and quality of qualitative research reporting.

The guidelines cover various aspects of qualitative research, including study design, data collection, analysis, and interpretation.

259: How many students was it pre-tested? How did they infer or understand to implement it to the participants, based on their feedback?

Line 381: 2.7 Discussion ( Not clear why it is here? ) in the result section

Line 510-512: Why not both?

Result section well explained

597: Discussion:

Must include characteristics about their population, clinical group vs general group. Common age group, gender, age.

(My doubt- if males then heavy or moderate, same for female)

- Did they consider cultural aspects of drinking in their sample, it can vary from culture to culture

- Why did the participants reduce or suggest reasons?

610-626 : Just listed their results, I feel add reasons to use those themes stated from the previous literature.

For eg: Family? What is mentioned in the literature (Good family support may help in abstinence and regular follow-ups) – and that’s why its deemed to be included in the their questions.

678- Attrition was high, due to the length of questionnaire (That could be one, the author must explain other factors too, as mentioned above)

- And if a questionnaire is long, it would pose a problem for future participants to adhere too?

Overall: Clearly elicit all abbreviations in all places.

Discussion can be improved, based on above suggestions.

Reviewer #7: 1. The claim that "no interventional GM study had been conducted on alcohol consumption [22]" is inaccurate and misleading. The reference provided, ref 22, is mentioned as anonymous, but it is incorrect to assume that the study itself is anonymous. In fact, there is a study conducted by Jan Klimas and others that specifically focuses on the effects of alcohol consumption.

2. A recent meta-analysis [36] showed that 16 studies used implementation intentions in relation to alcohol outcomes. II have been used to successfully reduce binge drinking [37], could you please clarify what this " II " in the begning of the sentence?

3. While the GMII approach is relatively new, there is limited research and empirical evidence supporting its effectiveness. More studies are needed to establish its long-term outcomes and generalizability across populations. The GMII intervention may not sufficiently account for the unique needs and circumstances of each individual struggling with alcohol misuse. As it follows a standardized approach, it may not address specific underlying issues that contribute to alcohol-related problems. Authors may want to talk about this that too.

4. The GMII intervention may not sufficiently account for the unique needs and circumstances of each individual struggling with alcohol misuse. As it follows a standardized approach, it may not address specific underlying issues that contribute to alcohol-related problems. It would be better authors could consider discuss the limitations of GMII intervention.

5. Participants from the general population were recruited using an online recruitment platform (i.e., the French Risc list). Research recruitment using an online platform offers several advantages, such as convenience, reach, and cost-effectiveness. However, it is essential for authors to acknowledge the biases, limitations, and potential problems associated with this method.

6. Selected material can be compared to the current literature. If no GM intervention targeting alcohol consumption existed to our knowledge [22]. Are there any studies in anmial model, please mention of those and discuss the relavance.

7. Overall, an excellent study and an important addition to current literature. The tile is great, the aim of the paper is clear and relevant. The introduction and background of the topic is very detailed and informative. Researchers have raised and outlined an important question relevant to the topic. No major flaws in the methodology used, no major flaws in the data presented, no misleading or false conclusions. The discussion section is detailed from multiple angles and placed in the context of the topic, It doesn't look under or overly interpreted. Just a few minor items to be improved as above

6. PLOS authors have the option to publish the peer review history of their article (what does this mean?). If published, this will include your full peer review and any attached files.

Reviewer #1: No

Reviewer #2: **Yes: **Aditi Sharma

Reviewer #3: No

Reviewer #4: No

Reviewer #5: **Yes: **Jasleen Kaur MD

Reviewer #6: No

Reviewer #7: No

---

## [Author Response · Author response to Decision Letter 0]

26 Dec 2023

Dear Editor,

We would like to thank you and the reviewers the opportunity to submit a revised version of our manuscript. The comments and suggestions were helpful, and undoubtedly helped to improve the quality of the document. We hope that the response we provide here is adequate. 

Our responses are in red under each suggestion, and lines number are indicated to refer modifications in the “clean copy” document.

Regards,

The authors

Reviewer #1: 

The authors have written a very thorough and well though out paper on the development of their intervention to address AUD with the help of GM and II and I really appreciate the opportunity to review this article. This paper may well have been two papers but the combination makes for an interesting read. I would recommend the following edits:

1. The authors should use IN the general population in line 47.

We used the word “among” as per reviewer 5 comment

2. The authors should explain the statistic 100 million in line 57 as to whether it is for country, continent or world.

This information has been added:

 “Around 1.3% of the world population (more than 100 million people) were diagnosed with an AUD in 2016”

3. In line 91, the authors should explain "4.3 reattribution".

The definition of this category has been added lines 114-117:

“From the behavior change taxonomy of Michie and collaborators [1], GM interventions would be categorized in the “shaping knowledge” cluster, and the technique “4.3 – reattribution” (i.e., elicit perceived causes of behavior and suggest alternative explanations, such as external or internal and stable or unstable).”

4. Overall, while using decimals the authors should use "." instead of ",".

It has been replaced in the entire manuscript.

5. The n for Study 1 groups seems low and the authors should explain why this was the size chosen for the study.

An explanation is provided line 258

“The number of groups was limited to four due to time, funding and recruitment constraints.”

6. The exclusion of hazardous drinking for Study 2 may skew the results of the study and the authors should explain why this was done.

 The hazardous drinking criteria was not used for study 2 in order to better represent the general population (i.e., not clinically diagnosed with AUD), by taking into account individuals with a potentially hazardous consumption. 

This is now explained lines 491-495 

“However, the hazardous drinking criteria for the general population was removed, meaning hazardous or dependent alcohol consumers (AUDIT score ≥ 7 for men and ≥ 6 for women) not diagnosed with AUD could participate in the study. Thus, the sample was more representative of the general population of drinkers, containing individuals with different degrees of severity in hazardous drinking.”

 7. The authors may want to explain why malleability was greater in the clinical group than general population

This was discussed lines 724-729:

 “This higher score for the clinical group might mean that individuals diagnosed with AUD and engaged in a therapeutic process were rightfully convinced that they can change and improve their alcohol consumption. Conversely, it could also mean that they lacked insight about their alcohol consumption, and failed to see the problem they had with alcohol. This would reflect a common discourse among patients with AUD, namely that they can stop drinking whenever they want and that they do not have an alcohol problem.”

8. The general discussion although useful summary seems redundant and repetitive and the authors may want to omit this section as it increases the length of the paper while not adding much value to it.

The discussion of study 2 and the first section of the general discussion were shortened to decrease the length of the paper and avoid redundancy.

The results of the validation study would make for an interesting read and I look forward to the authors publishing the results of the study.

Thank you for your comments

Reviewer #2: 

The article describes the creation of a psychosocial intervention called Growth Mindset and Implementation Intentions (GMII) to reduce alcohol consumption. The intervention was developed using a mixed method approach, combining qualitative and quantitative research methods. Focus groups were conducted to gather arguments supporting a malleable view of alcohol consumption and situations that trigger the desire to drink alcohol, along with strategies to counteract this need. The intervention consists of a popularized scientific article describing alcohol consumption as malleable, followed by two internalization exercises and a volitional help sheet to form plans. The document also discusses the added value of the intervention compared to existing tools and plans for future testing

The mixed-method study on the psychosocial intervention combining growth mindset and implementation intentions (GMII) to reduce alcohol consumption exhibits several noteworthy strengths. The research question is clearly articulated, and the study effectively conveys its purpose, focusing on the examination of GMII's impact on reducing alcohol consumption. A comprehensive literature review demonstrates a strong foundation in prior research. The mixed-methods approach, which combines quantitative and qualitative analyses, provides a multifaceted understanding of GMII's efficacy and its underlying mechanisms. The study utilizes sound data collection and analysis methods, ensuring research rigor. Furthermore, the inclusion of a diverse participant sample enhances external validity, and a thorough discussion of the practical implications of the findings illustrates the real-world relevance of the research.

The language used appears to be clear and concise. The information is presented in a straightforward manner, making it easy to understand the purpose and methodology of the study. The language is also technical and scientific, which is appropriate for a research article.

In terms of cohesiveness, the document seems to flow logically, with each section building upon the previous one. The study's objectives, methods, and findings are presented in a coherent manner, allowing the reader to follow the research process and understand the progression of the study.

In essence, this research article presents a compelling and excellently composed exploration of a novel treatment approach for addressing alcohol consumption. Its structural organization, encompassing a well-articulated introduction, substantive body, and a concise conclusion, notably facilitates readability and comprehension. What particularly stands out in this article is the depth of scholarship underpinning it. The author adeptly harnesses a wide array of sources, encompassing academic studies, scholarly papers, and expert viewpoints, to substantiate their arguments and findings. The extensive citation of sources further underscores the reliability and veracity of the presented information. In summation, this article represents a valuable and noteworthy addition to the academic domain. It imparts significant insights and poses pertinent questions that beckon further investigation.

We thank you for your feedback on our manuscript.

Reviewer #3: 

The manuscript is based on impressive empirical evidence and makes an original contribution but there should be some comment on possible bias like reporting bias of the study participants and their understanding of the difference between pre-clinical and actual alcohol use disorder etc.

Thank you for your comments

To improve the readability of the paper, I suggest that Manuscript is too long and should be shortened.

The discussion of study 2 and the first section of the general discussion were shortened to decrease the length of the paper and avoid repetitions.

Reviewer #4: 

Thank you for the opportunity to review this manuscript that describes a two part qualitative study that focused on development of a psychosocial intervention to reduce alcohol consumption. 

The manuscript was extremely detailed and the methods/results were well described. The methods and results section were written well and provided good information in an understandable way.

A few thoughts:

- There were a few grammatical errors throughout the manuscript, especially in the abstract and introduction sections (wrong use of past tense vs present, spelling,

Would recommend that authors revise the introduction section (which did have a good amount of detail) to address these issues.

The manuscript was edited for grammatical errors and other mistakes

- The behavioral change taxonomy part of the introduction was also hard to understand at times, especially when other behavioral change models were discussed. This section would benefit from some editing.

The taxonomy and models cited are now properly introduced and explained at the end of the 1.1 section (lines 83-99). It should now be easier to understand when they are referenced later in the manuscript.

“In this idea, Michie and al. [1] developed a taxonomy indexing 93 behavior change techniques, grouped under 16 clusters (e.g., goals and planning; reward and threat; self-belief; etc.). Furthermore, they introduced the behavior change wheel [14] to help identify and apply these techniques when targeting a specific change in individuals. At the center of the wheel is the COM-B model (Capability, Opportunity, Motivation – leading to Behavior). It posits that it is possible to encourage change by acting on the different sources of behavior, namely individuals' abilities (e.g., increasing knowledge about alcohol and its risks); opportunities (e.g., creating opportunities to carry out another activity instead of drinking alcohol); and motivation (e.g., encouraging the desire to change one's consumption). Another model, the transtheoretical model of change [15] complements the COM-B by proposing stages of change and strategies for moving through these stages. In the precontemplation stage, the individual doesn't think they have a problem with the product, and therefore doesn't want to change their behaviour. In the contemplation stage, they become aware of the problem but are still reluctant to change their consumption. In the preparation stage, they are motivated and ready to change their behaviour. In the action stage, the change in behaviour is effective. This is followed by the maintenance stage, in which the new behaviour is consolidated over time. Finally, there is the relapse stage, which is an integral part of the behavioural change process. The present project aimed to act on most sources of behavior and phases through the mindset theory and implementation intention strategies.”

- The authors explain the inclusion of both general population and those with alcohol use disorder (AUD) as reflecting diversity of alcohol use. While this help capture more data for the design of an intervention, does this potentially affect/dilute what the effect of the intervention can be on those with AUD?

 This is a good point and one of the main critiques that could be made of this intervention. By targeting all alcohol consumers, the material might be considered too “light” for individuals with AUD. The next study (testing the intervention among both population) will tell us if that is the case. 

This limitation was added in the Discussion section lines 736-739

“Relatedly, creating an intervention targeting both the general and clinical populations might have “diluted” the data, meaning that it could lack efficacy on the clinical population that present specific needs because it was also adapted to the general population.”

- Why was there a difference in the length of the focus groups between the populations?

An explication was added lines 271-277:

“The longer discussion time for the clinical population might be explained by the fact that clinical participants were more used to discuss in a group context. Indeed, some participants had already taken part in group therapy sessions at the time of recruitment. For the general population, participants might never have taken part in such a group discussion before, and were potentially less comfortable to share, or more reluctant to express personal ideas in public. This caused the exchanges to be shorter and resulted in shorter groups duration.”

- The small population is a limitation that the authors rightfully point out. Another limitation seem to be the characteristic of the sample with more women than men; which may not accurate reflect both the general population or those with AUD. This should be mentioned.

This limitation was added line 715-719:

“Furthermore, this work’s samples contained more females than males. This might not accurately represent the clinical or general populations of drinkers, still predominantly male [7]. Nonetheless, research show a recent increase in women’s alcohol consumption, despite sex and gender differences still existing regarding the trajectory of hazardous alcohol consumption, its clinical consequences, and social representations of alcohol [86].”

Overall good manuscript that can be improved with some changes.

We thank you for your comments

Reviewer #5: 

The article combining the growth mindset and implementation intentions to reduce alcohol consumption is beautifully written and the writer has been very candid in acknowledging that the study could have been helped better by increased sample size.

Thank you for this feedback. We have taken your comments into account, as detailed below.

There are a few arguments against the article though:

1. The study states that clinical population had a higher growth mindset than general population. What was the reason that people with AUD had stronger beliefs about malleability of alcohol consumption. It could also mean that people who had AUD got to that stage because of lack of insight and failing to acknowledge that alcohol is a big problem for them. A lot of clients with AUD continue to say they know how to stop drinking and they can do it anytime contrary to the fact they are the ones fulfilling AUD criteria.

SOLUTION: Please include in limitation that we don't know the reasons why clinical population had a higher growth mindset than general population.

This limitation was included in the discussion lines 724-729 and two explanations are provided:

 “This higher score for the clinical group might mean that individuals diagnosed with AUD and engaged in a therapeutic process were rightfully convinced that they can change and improve their alcohol consumption. Conversely, it could also mean that they lacked insight about their alcohol consumption, and failed to see the problem they had with alcohol. This would reflect a common discourse among patients with AUD, namely that they can stop drinking whenever they want and that they do not have an alcohol problem.”

2. The biological underpinnings of drinking cannot be ignored. The study focuses on attitudes and responses of alcohol drinkers but ignores the genetics, heritability of alcohol use disorder which is a well documented phenomenon. So grouping everyone with AUD in one broad group with disregard for genetics, trauma, ethnicity is an over simplistic approach.

SOLUTION: Please include the fact in limitation of study that this intervention could be very simplistic in the complex and multifactorial world of alcohol dependence.

This limitation was included in the discussion lines 730-736. Please note that the project did not aim to construct a tool that would be sufficient to treat alcohol disorders, but that address two determinants of alcohol consumption:

 “The constructed intervention focused on changing two determinants of alcohol consumption: beliefs about alcohol, and behavioral responses of individuals in relation to their consumption. Alcohol abuse is a complex phenomenon, with physiological (e.g., genetics, heritability, ethnicity) as well as socio-cultural (e.g., past traumas, education, socio-economic status) factors at play. Consequently, it is important to note that the intervention did not aim to address all facets of alcohol consumption, but only these two determinants. This focus on beliefs and behavioral responses might help inform potential obstacles to the success of existing prevention programs and therapy.”

3. The tools described in the article can be used as coping skills for maintaining sobriety but are not enough by themselves. Alcohol use is a very complex disorder with intrinsic and extrinsic locus of controls. Focusing on the fact that changing the viewpoint about malleability of drinking and coming up with appropriate responses in critical situations will go a long way in staying abstinent for people with alcoholism. However it ignores the fact that such simplistic approach can backfire. Failure to stay abstinent despite using these techniques can lead to shame and guilt and further reclusiveness and hopelessness on part of alcoholics.

SOLUTION: This technique can perhaps be an important coping skill to teach people to maintain sobriety but has to be used in conjunction with other proven methods like AA, having a sponsor and alcohol cessation medications.

This limitation was included in the discussion lines 739-744

“Finally, failure to reach one’s desired objectives (abstinence from alcohol or reduced drinking) following the intervention might cause a blowback and lead to further discouragement and guilt from people with AUD. Without tests of the efficacy of the intervention in the general and clinical populations, the tool should be used in conjunction with other proven methods such as support groups, therapy, or medication. All these limitations will be the subject of special attention in the test of the efficacy of the intervention.”

MINOR CORRECTION:

LINE 47 of patients with Alcohol Use Disorder (AUD) and alcohol users the general population. The intervention

PLEASE INCLUDE alcohol users AMONG the general population

 Those modifications have been made

Reviewer #6: 

Summary of the Paper:

The paper aims to create a psychosocial intervention combining growth mindset and implementation intentions (GMII) to reduce alcohol consumption among both the general population and individuals with alcohol use disorder. The intervention consists of a popularized scientific article describing alcohol consumption as malleable, followed by internalization exercises and a volitional help sheet with situations and solutions. The study used a mixed method approach, including qualitative focus groups to extract arguments and strategies related to alcohol consumption, and quantitative research to create a questionnaire scoring the relevance of each argument, situation, and strategy. The intervention was developed based on the highest scored arguments, situations, and strategies. The paper discusses the added value of the intervention compared to existing tools and plans to test it in future studies

Aim and Objective:

The aim of this study is to create a psychosocial intervention combining growth mindset and implementation intentions (GMII) to reduce alcohol consumption among both the general population and individuals with alcohol use disorder. The objective is to develop an intervention that includes a popularized scientific article describing alcohol consumption as malleable, internalization exercises, and a volitional help sheet with situations and solutions. The study uses a mixed method approach, including qualitative focus groups to extract arguments and strategies related to alcohol consumption, and quantitative research to create a questionnaire scoring the relevance of each argument, situation, and strategy. The intervention is based on the highest scored arguments, situations, and strategies, and the paper discusses its added value compared to existing tools and plans to test it in future studies.

Line 71: AIM mentioned. (The objective didn’t stand out clearly)

The aims of the study was moved earlier (in the 1.1 section), and made clearer lines 76-81

“The first aim of the present project was then to construct a single intervention able to foster changes in drinking behavior among different types of consumers (from occasional drinkers to individuals with AUD), by targeting two processes thought to be common among them: implicit theories of alcohol consumption, and consumption automatisms.

 The second aim of the project was to implement empirically tested and complementary behaviour change strategies.”

Line 127: Abbreviation of Implementation Intentions not clearely mentioned- directly mentioned II

Abbreviation is now clearly mentioned line 152

Line 179: Purpose

Targeting Both General and Clinical Populations

The researchers aimed to design a psychosocial intervention to reduce alcohol consumption among both the general population of drinkers and individuals with Alcohol Use Disorder (AUD).

This was done to reflect the diversity of individuals affected by alcohol consumption, with varying degrees of severity.

Implicit beliefs about alcohol consumption, as well as situations and strategies pertinent to users, could greatly vary depending on their profile.

By focusing on both the clinical and general population, the intervention would account for this diversity.

Mixed Approach Adopted

The lack of literature for Growth Mindset Interventions (GMIs) on substance and alcohol use disorders, as well as the specificity of the targeted populations, led the researchers to adopt a mixed approach.

The mixed approach involved using empirical data in addition to the scientific literature to design the intervention.

Firstly, in study 1, with a qualitative focus, focus groups were conducted with reflexive thematic analysis to extract arguments promoting a malleable view of alcohol consumption (study 1A).

Secondly, in study 1B, critical situations that might arouse the needs, desires, or habits to drink alcohol, and appropriate responses that people might use in those situations to not drink or drink less were identified.

Thirdly, in study 2, adopting a quantitative method, each argument, situation, and strategy collected during the focus groups was scored.

The best-ranked items, as well as input from the scientific literature, were finally used to create the GMII intervention.

Study Samples

As the intervention's objective is to reduce alcohol consumption among the general and clinical populations of alcohol consumers, each study's sample was taken from both populations.

The mixed method approach combined qualitative and quantitative research methods among both populations.

The data was analyzed using reflective thematic analysis in line with the scientific literature on alcohol consumption.

Line 201: SRQR guidelines refer to the Standards for Reporting Qualitative Research. (Only abbreviation mentioned)

The SRQR meaning is now mentioned line 227

These guidelines provide a framework for reporting qualitative research studies.

They aim to improve the transparency, rigor, and quality of qualitative research reporting.

The guidelines cover various aspects of qualitative research, including study design, data collection, analysis, and interpretation.

259: How many students was it pre-tested? How did they infer or understand to implement it to the participants, based on their feedback?

Information has been added regarding the pretest and the use of pre-test feedbacks was described lines 295-299

“For study 1, two semi-structured discussion guides were used to conduct the focus groups. The discussion guides were pretested on a sample of university students (2 groups, n1 = 9, n2 = 8) to design questions and probes. The main questions were asked to students during the pretest, and the variety of their response helped to refine the probes to adequately cover the questions’ subjects and elicit pertinent data, as well as improve their understandability.”

Line 381: 2.7 Discussion (Not clear why it is here?) in the result section

This is the discussion section of studies 1A and 1B only, before moving to study 2.

Line 510-512: Why not both?

Please see the introduction of the concept lines 101-104. According to the theory, one’s mindset about a particular trait (here alcohol consumption) is positioned on a continuum between “fixed” and “growth”. While a mindset can evolve along this continuum in time, and while individuals can hold several fixed and growth mindsets about different attributes (e.g., growth mindset of intelligence, but fixed mindset of personality), a specific mindset, measured at one point in time, would either be fixed or growth.

Result section well explained

597: Discussion:

Must include characteristics about their population, clinical group vs general group. Common age group, gender, age.

(My doubt- if males then heavy or moderate, same for female)

A table describing participants’ characteristics was added line 259 for clarity. Participants means audit score by sex for the clinical groups were added lines 249-256. Furthermore, it was indicated that all individual scores were higher than the cutoff of 13 suggested for alcohol dependence in the French population.

 “Mean AUDIT scores in the clinical group were M = 25.6, SD = 6.5 for women and M = 18.0, SD = 1 for men. For study 1B (N = 12), five participants were recruited from the general population (three women and two men) to form the first group. Ages ranged from 32 to 80 years old (M = 54, SD = 21.1, one participant did not report their age). Seven participants were recruited from the clinical population (two women and four men) to form the second group, ages ranged from 31 to 72 years old (M = 49.4, SD = 12.7). Mean AUDIT scores in the clinical group were M = 30.7, SD = 2.52 for women and M = 23, SD = 10.9 for men. All individual scores were higher than the cutoff of 13 suggested for alcohol dependence in the French population [52].”

An analysis of differences of AUDIT score by sex and group (clinical vs. general) was added for study 2 lines 557-561. The effect was not significant.

“A general linear model analysis was used to assess potential differences in AUDIT scores between men and women for the general and clinical populations F(1,93) = 1.04, p = .31. The interaction was not significant. However, AUDIT scores significantly differed between groups, F(1,93) = 98.94, p < .001, η²p = .51, the clinical population scoring higher (M = 25.05, SD = 6.51) than the general population (M = 9.03, SD = 7.64), as might have been expected.”

- Did they consider cultural aspects of drinking in their sample, it can vary from culture to culture

The influence of culture on drinking was mentioned during the groups in “social norms of alcohol” in group 1A and “culture of alcohol” component in group 1B. A paragraph was also added in the limitation section line 629-635:

 “If the qualitative data collected was rich, it could nonetheless be hypothesized that more themes reflecting diverse beliefs, situations and responses related to alcohol consumption might have emerged with more groups. Cultural aspects of drinking, for example, might influence beliefs and subsequent drinking [64]. If culture was mentioned by participants (“social norms of alcohol” component in group 1A and “culture of alcohol” component in the group 1B), more groups would have allowed further room for cultural differences to be expressed and integrated into the intervention.”

- Why did the participants reduce or suggest reasons?

Participants reduce alcohol for various reasons such as deleterious health of daily life consequences. They can be found in the “reasons to manage” subtheme in table 4 and lines 397-402:

“Managing alcohol consumption. The second theme focused on the reasons and the way participants manage their alcohol consumption. Participants invoked several reasons to manage their alcohol consumption. They fear the potential consequences of alcohol on their health (e.g., diseases, weight gain) and daily life (e.g., negative consequences with family and friends, being unable to function well the next day), and might have experienced a moment of realization driving them to change, be it by specific events (e.g., car crash) or relatives’ intervention.”

610-626: Just listed their results, I feel add reasons to use those themes stated from the previous literature.

For eg: Family? What is mentioned in the literature (Good family support may help in abstinence and regular follow-ups) – and that’s why its deemed to be included in the their questions.

This was done in study 1 discussion section lines 421-464. The themes found and their pertinence to the intervention are linked to what can be found in the scientific literature.

Study 2’s discussion and the general discussion were shortened to avoid redundancy and listing previous results.

678- Attrition was high, due to the length of questionnaire (That could be one, the author must explain other factors too, as mentioned above)

This limitation was further detailed lines 640-645:

“Attrition was high, probably due to the length of the two-part questionnaire. Indeed, study 2 was done online and autonomously by participants. Despite the 20€ reward incentive to complete the study, the remote and dense nature of the task might have discouraged engagement. Technical issues preventing access to the questionnaire (e.g., internet connection, email errors) might also have occurred. A potential solution would have been to reduce the questionnaire’s length, or to invite participants to do the study in person in a laboratory context.”

- And if a questionnaire is long, it would pose a problem for future participants to adhere too?

The questionnaire was long due to the quantity of arguments (N = 27), situations (N = 62) and responses (N = 47) needing to be evaluated. In the final intervention, only 22 arguments, 20 situations and 20 responses were included. The whole intervention is 5 pages long for approximately 30 minutes of completion time, which was deemed adequate. The material will be tested in a future study, and feedback from participants will be collected on length and engagement. 

Overall: Clearly elicit all abbreviations in all places.

Discussion can be improved, based on above suggestions.

Thank you for your comments

Reviewer #7: 

1. The claim that "no interventional GM study had been conducted on alcohol consumption [22]" is inaccurate and misleading. The reference provided, ref 22, is mentioned as anonymous, but it is incorrect to assume that the study itself is anonymous. In fact, there is a study conducted by Jan Klimas and others that specifically focuses on the effects of alcohol consumption.

The reference is marked “anonymous” to preserve the anonymity of the present manuscript. This anonymous review showed that no GM intervention was made on alcohol consumption. In any case, we specifically mentioned interventional studies targeting alcohol outcomes using a growth mindset intervention. According to previous research, none existed at the time of this study. 

We removed the GM abbreviation for clarity lines 135-136:

“To our knowledge, no interventional growth mindset study had been conducted on alcohol consumption at the time of this study [22]”

2. A recent meta-analysis [36] showed that 16 studies used implementation intentions in relation to alcohol outcomes. II have been used to successfully reduce binge drinking [37], could you please clarify what this " II " in the beginning of the sentence?

This was clarified. II means “Implementation intentions”

3. While the GMII approach is relatively new, there is limited research and empirical evidence supporting its effectiveness. More studies are needed to establish its long-term outcomes and generalizability across populations. The GMII intervention may not sufficiently account for the unique needs and circumstances of each individual struggling with alcohol misuse. As it follows a standardized approach, it may not address specific underlying issues that contribute to alcohol-related problems. Authors may want to talk about this that too.

Please find a response below

4. The GMII intervention may not sufficiently account for the unique needs and circumstances of each individual struggling with alcohol misuse. As it follows a standardized approach, it may not address specific underlying issues that contribute to alcohol-related problems. It would be better authors could consider discuss the limitations of GMII intervention.

As per reviewer 4 and 5 comments, this has been discussed lines 730-739

“The constructed intervention focused on changing two determinants of alcohol consumption: beliefs about alcohol, and behavioral responses of individuals in relation to their consumption. Alcohol abuse is a complex phenomenon, with physiological (e.g., genetics, heritability, ethnicity) as well as socio-cultural (e.g., past traumas, education, socio-economic status) factors at play. Consequently, it is important to note that the intervention did not aim to address all facets of alcohol consumption, but only these two determinants. This focus on beliefs and behavioral responses might help inform potential obstacles to the success of existing prevention programs and therapy. Relatedly, creating an intervention targeting both the general and clinical populations might have “diluted” the data, meaning that it could lack efficacy on the clinical population that present specific needs because it was also adapted to the general population.”

5. Participants from the general population were recruited using an online recruitment platform (i.e., the French Risc list). Research recruitment using an online platform offers several advantages, such as convenience, reach, and cost-effectiveness. However, it is essential for authors to acknowledge the biases, limitations, and potential problems associated with this method.

This was discussed lines 710-714

“The recruitment platform used for the general population (i.e., the Risc list) presented several advantages such as convenience, reach, and cost-effectiveness. However, it also carried potential limitations and biases, such as a non-representative population (as it is originally an academic platform, a lot of participants are students), and an incentive to participate due to the financial reward, which might have negatively impacted participants’ engagement and responses quality.”

6. Selected material can be compared to the current literature. If no GM intervention targeting alcohol consumption existed to our knowledge [22]. Are there any studies in anmial model, please mention of those and discuss the relavance.

 To our knowledge, no GM or II intervention exist in the animal model. The nature of these GM and II interventions implies higher cognitive functions (i.e., beliefs about the malleability of personal traits, goal planning) characteristics to humans.

7. Overall, an excellent study and an important addition to current literature. The tile is great, the aim of the paper is clear and relevant. The introduction and background of the topic is very detailed and informative. Researchers have raised and outlined an important question relevant to the topic. No major flaws in the methodology used, no major flaws in the data presented, no misleading or false conclusions. The discussion section is detailed from multiple angles and placed in the context of the topic, It doesn't look under or overly interpreted. Just a few minor items to be improved as above

We thank you for your feedback.

---

## [Decision Letter · Decision Letter 1]

10 Jan 2024

Creating a psychosocial intervention combining growth mindset and implementation intentions (GMII) to reduce alcohol consumption: A mixed method approach

PONE-D-23-18196R1

Dear Dr. Parada,

We’re pleased to inform you that your manuscript has been judged scientifically suitable for publication and will be formally accepted for publication once it meets all outstanding technical requirements.

Kind regards,

Lakshit Jain, MD

Academic Editor

PLOS ONE

Additional Editor Comments (optional):

Reviewers' comments:

Reviewer's Responses to Questions

**Comments to the Author**

1. If the authors have adequately addressed your comments raised in a previous round of review and you feel that this manuscript is now acceptable for publication, you may indicate that here to bypass the “Comments to the Author” section, enter your conflict of interest statement in the “Confidential to Editor” section, and submit your "Accept" recommendation.

Reviewer #1: All comments have been addressed

Reviewer #4: All comments have been addressed

Reviewer #5: All comments have been addressed

Reviewer #8: (No Response)

2. Is the manuscript technically sound, and do the data support the conclusions?

Reviewer #1: Yes

Reviewer #4: Yes

Reviewer #5: Yes

Reviewer #8: Yes

3. Has the statistical analysis been performed appropriately and rigorously? 

Reviewer #1: Yes

Reviewer #4: Yes

Reviewer #5: Yes

Reviewer #8: Yes

4. Have the authors made all data underlying the findings in their manuscript fully available?

Reviewer #1: Yes

Reviewer #4: No

Reviewer #5: Yes

Reviewer #8: Yes

5. Is the manuscript presented in an intelligible fashion and written in standard English?

Reviewer #1: Yes

Reviewer #4: Yes

Reviewer #5: Yes

Reviewer #8: Yes

6. Review Comments to the Author

Reviewer #1: The authors have thoroughly addressed all reviewer comments and the paper is much more thorough, detailed and yet concise and currently appear fitting to be published in the journal.

Reviewer #4: The authors have satisfactorily addressed all my comments.

The introduction section is improved especially with the addition of the detail around behavioral change models.

Thank you for the work on this manuscript.

Reviewer #5: The authors have addressed all the concerns that were raised during the review. The discussion section now addresses all the shortcomings that were pointed out initially.

Reviewer #8: Dear Dr. Lakshit Jain MD

I have had the honor of reviewing the manuscript titled "Creating a psychosocial intervention combining growth mindset and implementation intentions (GMII) to reduce alcohol consumption: A mixed method approach” submitted to PLOS ONE for peer review.

The manuscript addresses the development of a novel psychosocial intervention aimed at reducing alcohol consumption. The study combines growth mindset theory and implementation intention strategies through a mixed-methods approach involving focus groups and quantitative research among both general and clinical populations.

The strengths of this manuscript lie in its comprehensive approach to intervention development. The use of focus groups to extract arguments, situations, and strategies, followed by a quantitative analysis to score their relevance, is a robust methodology. The authors successfully integrate diverse themes and perspectives related to alcohol consumption, providing a well-rounded foundation for the intervention. The comparison of the created material with existing literature on growth mindset interventions targeting other substances adds valuable context and highlights the uniqueness of the developed GMII intervention.

However, there are some limitations to consider. The sample size in both focus groups and the subsequent questionnaire study is relatively small, and the high attrition rate in the online questionnaire study may affect the generalizability of the findings. Additionally, the manuscript acknowledges the possibility of cultural differences influencing beliefs about alcohol consumption, which may not be fully captured with the current study's sample. Additional limitations included as the authors acknowledged such as the focus on specific determinants of alcohol consumption and the potential "dilution" of data due to targeting both general and clinical populations.

7. PLOS authors have the option to publish the peer review history of their article (what does this mean?). If published, this will include your full peer review and any attached files.

Reviewer #1: No

Reviewer #4: No

Reviewer #5: **Yes: **Jasleen Kaur

Reviewer #8: No

---

## [Editor Report · Acceptance letter]

23 Jan 2024

PONE-D-23-18196R1 

PLOS ONE

Dear Dr. Parada, 

I'm pleased to inform you that your manuscript has been deemed suitable for publication in PLOS ONE. Congratulations! Your manuscript is now being handed over to our production team.

Kind regards, 

on behalf of

Dr. Lakshit Jain 

Academic Editor

PLOS ONE